# Localized shear versus distributed strain accumulation as shear-accommodation mechanisms in crustal shear zones: Constraining their dictating factors

Pramit Chatterjee[1], Arnab Roy[1], and Nibir Mandal[1]

[1]Department of Geological Sciences, Jadavpur University, Kolkata 700032, India

**Correspondence:** Nibir Mandal (nibir.mandal@jadavpuruniversity.in)

**Abstract.** Understanding the underlying mechanisms of strain localization in Earth's lithosphere is crucial to explain the mechanics of tectonic plate boundaries and various failure-assisted geophysical phenomena, such as earthquakes. Geological field observations suggest that shear zones are the most important lithospheric structures of intense shear localization at the plate boundaries to share a major part of tectonic deformations. Despite extensive studies in the past several decades, the factors governing how the shear zones accommodate the bulk shear, whether by distributed homogeneous strain (i.e., development of macroscopic S foliations normal to the principal shortening strain axis) or by localized shearing (formation of shear-parallel C bands) remain largely unexplored. This study aims to address this gap in knowledge, providing observational evidences of varying S and C development in crustal shear zones from two geological terrains of Eastern India. The field observations are complemented with 2D-viscoplastic numerical simulations within a strain-softening rheological framework to constrain the factors controlling the two competing shear-accommodation mechanisms: homogeneously distributed strain accumulation versus shear band formation. The model-based analysis recognizes the bulk shear rate ($\dot{\gamma}_b$), the initial viscosity ($\eta_v$) and the initial cohesion ($C_i$) of a shear zone as the most critical factors to determine the dominance of one mechanism over the other. For a given $C_i$, low $\dot{\gamma}_b$ and $\eta_v$ facilitate the formation of S foliation (uniformly distributed strain), which transforms to C-dominated shear-accommodation mechanism with increasing $\eta_v$. However, increasing $\dot{\gamma}_b$, facilitates shear accommodation in a combination of the two mechanisms leading to CS- structures. The article finally discusses the conditions in which shear zones can enormously intensify localized shear rates to produce rapid slip events, such as frictional melting and seismic activities.

**Keywords**: deformation localization, shear accommodation mechanism, field analysis of shear structures, finite element modelling, viscoplastic rheology, rheological weakening and slip events

# 1 Introduction

Shear zones are long, narrow regions of strain localization, relative to their surroundings, that accommodate large amounts of tectonic movements. They occur on varied scales, ranging from grain (millimetres) to crustal scales (hundreds of kilometres) and at varying depths, covering upper crust to the upper mantle region (Adam et al., 2005; Vauchez et al., 2012; Fossen and Cavalcante, 2017). In Earth's lithospheric deformations large-scale shear zones often play a critical role in triggering catastrophic phenomena, such as fault-driven earthquakes (Fagereng et al., 2014; French and Condit, 2019; Kotowski and Behr, 2019; Beall et al., 2021; Rodriguez Padilla et al., 2022), landslides (Korup et al., 2007; Hughes et al., 2020) and abrupt topographic modifications (Malik et al., 2006; Wang et al., 2020; Rodriguez Padilla, 2023). They also act as potential locations for shear-induced partial melting of rocks, as widely documented from their association with pseudotachylytes (Sibson, 1975; Papa et al., 2023), which can dramatically augment fault-slip rates and associated strain accumulation processes, leading to mega-earthquake events (Di Toro et al., 2006; Rice, 2006; Menegon et al., 2021). Understanding their internal shear-accommodation mechanisms has thus set a critical area of study in solid earth geophysics, especially with a special focus on the rheological contributions to shear enhancement processes. Both field and experimental observations suggest that crustal shear zones generally accommodate strain by contrasting micro-scale deformation mechanisms in brittle and viscous regimes (Fossen and Cavalcante, 2017), where grain scale fracturing, grain rotation and frictional sliding are the principal mechanisms in the brittle regime(Sibson, 1977; Logan, 1979), whereas dislocation-assisted creep (crystal-plastic), grain-boundary sliding and syn-kinematic recrystallization dominating mechanisms in the viscous regime (Passchier and Trouw, 2005). However, shear zones typically evolve through strain partitioning along macroscopic shear bands, irrespective of their internal deformation mechanisms, the growth of which is generally controlled by various microscale deformation mechanisms, such as, dynamic recrystallization syn-kinematic recrystallization, grain size reduction and fluid assisted mechanical weakening. The bands share a large fraction of the bulk shear in the shear zones.

The origin of shear bands in rocks and practical solid materials, such as metals and polymers has remained a subject of challenging studies over several decades, particularly in the context of failure analysis (Bowden and Raha, 1970; Wang and Lade, 2001; Torki and Benzerga, 2018; Finch et al., 2020; del Castillo et al., 2021). Compression test experiments on homogeneous isotropic solids show shear band localization in conjugate sets, with their dihedral angles varying in the range of 60° to 90°, depending on deformation conditions, such as the strain rate, and mechanical properties, such as coefficient of internal friction, dilatancy factor, strain hardening parameter(Bowden and Raha, 1970; Roscoe, 1970; Rudnicki and Rice, 1975; Vardoulakis et al., 1978; Anand and Spitzig, 1980, 1982; Wang and Lade, 2001; Kaus, 2010; Torki and Benzerga, 2018; Mukhopadhyay et al., 2023). Several theoretical models have predicted the shear band angles to the compression axis in isotropic materials as a function of the physical variables mentioned above (Hutchinson and Tvergaard, 1981; Anand and Su, 2005). Brittle-viscous layered composites also undergo failure in conjugate shear bands, although their modes of development switching from localized to distributed, with increasing brittle to viscous layer-thickness and viscosity ratios (Schueller et al., 2010). A similar line of failure studies focuses on the mechanisms of shear localization in simple shear deformation of granular rocks, which suggest that shear bands develop in a more complex manner, forming multiple sets, compared to a simple conjugate set of band for-

mation under compressional deformations. The multiple shear bands in granular materials, described as Y-, B-, P- and R bands form at characteristic angles to the bulk shear direction (Logan, 1979; Logan et al., 1992) , where Y and B bands are oriented parallel to the shear direction (B bands localize preferentially at the shear zone boundaries), and P bands occur at an angle of 15°- 45°with their vergence in the shear direction. R (Riedel) constitutes the most dominant two sets of antithetically vergent shear bands, one set at low-angles ($\sim 15° - 20°$) and the other at high angles (60°- 70°) to the shear direction, conventionally symbolized as $R_1$ and $R_2$ bands, respectively (Roy et al., 2021). In shear deformations these secondary shear bands generally occur as discrete planar zones, often marked by localization of gouge materials with intense grain-size reduction (Volpe et al., 2022; Casas et al., 2023).

The mechanics of shear band formation in shear deformation is still a lively problem, which has rejuvenated fresh theoretical and experimental studies of shear failure in the last couple of decades (Fossen, 2010; Hall, 2013). Numerical shear experiments on granular materials suggest that shear bands localize shear not by any bifurcation of local mechanical states, but by a long-range geometrical interaction of material particles (Ord et al., 2007). On the other hand, Mair and Abe (2008) have demonstrated from 3D numerical simulations a direct correlation of strain localization with grain size reduction in fault gouge. Laboratory experiments have been conducted on quartz-feldspar-rich granular materials and carbonates (Logan, 1979; Marone and Scholz, 1989; Marone et al., 1990; Beeler et al., 1996). A direction of these experimental investigations suggests that the relative growth of multiple sets of bands depends significantly on phyllosilicate and water content in granular aggregates (Morgenstern and Tchalenko, 1967; Wijeyesekera and De Freitas, 1976; Maltman, 1977; Logan and Rauenzahn, 1987; Rutter et al., 1986; Logan et al., 1992; Saffer and Marone, 2003; Collettini et al., 2011; Haines et al., 2013; Giorgetti et al., 2015; Orellana et al., 2018; Okamoto et al., 2019; Ruggieri et al., 2021; Volpe et al., 2022). Shear deformation of viscous materials also produce secondary shear bands, as in brittle materials, and their studies have gained serious attention due to their implications in interpreting various geodynamic processes, such as lithospheric subduction, deformation-assisted fluid/melts migration and earthquake generation in viscous regimes (Katz et al., 2006; Kirkpatrick et al., 2021; Beall et al., 2021; Tulley et al., 2022; Mancktelow et al., 2022). The mode of strain accommodation in viscous shear zones, however, largely differ from those discussed above for granular shear zones showing brittle behaviour. Although viscous materials typically accommodate shear by continuous deformations without any macroscopic fractures in shear zones (Rutter et al., 1986), many authors have reported brittle features such as riedel fractures, V-pull aparts in minerals,strain dependent transition from brittle fractures to mylonitic shearing, etc from viscous shear zones (Paterson and Wong, 2005; Fusseis et al., 2006; Fusseis and Handy, 2008; Mukherjee and Koyi, 2010; Doglioni et al., 2011; Meyer et al., 2017). Experiments performed in conditions to mimic the brittle-viscous transition zone show the localization of shear in the form of bands where both brittle and viscous features dominate as functions of temperature, strain rate and microscale mechanisms of deformation (Schmocker et al., 2003; Pec et al., 2016; Marti et al., 2017, 2018, 2020). Previous studies have used various rheological models to investigate the processes of strain localization in the lithosphere and their controlling factors. A direction of these studies have invoked a power-law viscous rheology to show the mechanism of shear localization in terms of porosity bands in viscous regimes (Katz et al., 2006). A parallel line of studies have combined a pressure dependent yield criterion with viscous rheology to deal with the problem of shear band formation (Mancktelow, 2006; Moresi et al., 2007a, b; Lemiale et al., 2008). On the other hand, Bercovici and

Karato (2002) have shown theoretically strain localization in lithosphere, taking into account thermal, damage, and grain-size controlled feedback mechanisms.

Extensive numerical and experimental modelling (Shimamoto, 1986, 1989; Burlini and Bruhn, 2005; Misra et al., 2009; Meyer et al., 2017; Finch et al., 2020) as well as field observations lead to a common finding that viscous shear zones accommodate their bulk shear deformation in two principal mechanisms: *uniformly distributed strain accumulation* and *localized shearing*. Distributed strain accumulation imparts pervasive planar fabrics (called S foliation in literature) tracking the XY plane of the finite strain ellipsoid, often defined by flattened grain shape and preferred orientations of phyllosilicates, e.g., muscovite, biotite, and chlorite. In contrast, localized shearing occurs in spaced zones, forming shear bands either parallel or at low angles to the principal shear plane, commonly described as C and C' bands, respectively (Berthé et al., 1979; Bos and Spiers, 2001; Niemeijer and Spiers, 2005, 2006; Tesei et al., 2012, 2014). These bands accommodate large shear strains, compared to the surroundings, and they are characterized by extreme grain refinement. In some cases, shear bands (termed as C'' bands) occur sporadically at high angles to the shear direction. Among these bands, C occurs as the most dominant structures in natural shear zones, and they develop as closely spaced planar zones to develop a foliation, as widely reported from typical SC mylonites in viscous shear zones, where the S and C foliations interact with one another, giving an anastomosing network structural characteristics in the sheared rocks. Some authors have described these fabrics also from brittle shear zones (Lin, 1999).

It follows from the preceding discussion that shear deformation in crustal shear zones generally occurs by a combination of distributed viscous strain (homogeneous S foliation development) and localized zones (shear band formation) of viscous strain (Lister and Snoke, 1984; Burlini and Bruhn, 2005; Mancktelow, 2006; Misra et al., 2009; Marques et al., 2011b, a), although their relative formation is still a subject of debate. Some authors (Berthé et al., 1979) have proposed that S-foliations and C- bands form simultaneously. However, recent studies (Bukovska et al., 2013; Bukovská et al., 2016) have claimed a non-synchronous origin of the two shear zone features, suggesting a significant time gap ($\sim$10 Myr) between the S folations and C band formation. On the other hand, some shear zones develop distributed viscous strains to produce penetrative planar fabrics, with little or no shear localization (Ramsay et al., 1983; Marques et al., 2013; Fossen and Cavalcante, 2017; Gomez-Rivas et al., 2017; Pennacchioni and Mancktelow, 2018), as often documented from shear zones with S mylonites, whereas another class of shear zones accommodates shear mainly by shear band with little or no distributed strain in the inter-band regions, as evident from the absence or little penetrative foliation development, even in the grain scale (Lister and Snoke, 1984; Mukhopadhyay and Deb, 1995; Lloyd and Kendall, 2005). This type of shear zones are generally dominated by C mylonites. What controls these two modes of shear accommodation is, however, less explored. In a recent study, Tokle et al. (2023) have addressed this problem from sheared quartzite, considering phyllosilicate content as a controlling factor, where phyllosilicates allow strains to localize preferentially in bands, leaving quartzite grains less deformed. Schueller et al. (2010) recognized composite structure of alternate low viscous and high viscous layers and their viscosity ratio as factors to determine the distributed versus localized fracturing in shear zones. Numerical simulations have shown that the growth of macro shear bands or mesoscopic scale slip planes without any macroscopic localized bands in granular materials can form, depending on initial densities and loading paths (Darve et al., 2021). Despite these studies, the problem of distributed viscous strain versus localized shear band

formation, especially in terms of a generalized rheological scheme needs further attention, which constitutes the central theme of this article.

To address this problem, the present study examines the modes of shear accommodation (distributed strain accumulation versus localized shearing) on macroscopic scales in crustal shear zones from the Chotonagpur Gneissic Complex and the Singhbhum Shear Zone, East Indian cratons. The shear zones show spectacular variations in their macroscopic structural features, based on which they are classified into three categories: i) shear zones dominated by shear-parallel high-strain zones (C bands), separated by regions of weak or no distributed deformation (S foliations), ii) shear zones dominated by S foliations, without any strong shear-parallel C band localization , and iii) shear zones with competing development of C bands and domains of distributed S foliations. We use visco-plastic models within a macroscopic rheological framework (continuum mechanics approach), where the viscous component is to simulate macroscopically distributed continuous deformations, while a yield behaviour (plastic component) is introduced to couple the process of strain localization in the viscously deforming shear zones. This rheological model allows to investigate the factors controlling these two competing mechanisms of deformation accommodation: distributed strain versus localized shear band formation. The article presents a map showing the fields of their growth as a function of two fundamental kinematic and rheological parameters: bulk shear rate and initial viscosity of shear zone rocks, respectively. This study also discusses possible shear-rate enhancement processes and their implications in underpinning the origin of slip-induced catastrophic processes, such as frictional melting and earthquakes in viscous regimes.

## 2   Field Observations

### 2.1   Study Area

We studied crustal shear zones in two tectonic regions of the Precambrian Craton: Singhbhum Shear zone (SSZ) and Chotanagpur Granite Gneissic Complex (CGGC) in Eastern India (Fig.1). A detailed description of their overall geological setting is presented in Supplementary (S1). The SSZ is a spectacular arcuate, about 200 km long and 2 km wide, thrust-type shear zone at the interface between the Archean nucleus on south and the North Singhbhum Mobile belt (NSMB). Our field investigations in the SSZ concentrated in its south-eastern flank at Patheragora village (22°32′37.911″N, 86°26′31.223″E), near the old Surda copper mines and Musabani (22°30′59.3″N 86°26′26.5″E) town in Purbi Singhbhum district, Jharkhand. The main rock types of this area are quartzite mylonites, mica and chlorite schists, and mylonitised granite. The CGGC lies north of NSMB, covered mostly by a variety of granite gneisses, dotted with minor lithologies, e.g., mafic and ultramafic intrusives (Mahadevan, 1992). The host rocks are metamorphosed to amphibolite to granulite grades (Roy et al., 2021). We conducted our field investigations in the northern part of Purulia District at Bero hills (23°31′54.4″N 86°45′35.5″E) and Belamu Pahar in Anandanagar (23°27′56.1″N 86°03′26.6″E), where excellent outcrop-scale viscous shear zones are exposed in granite gneisses. They are typically a few centimeters to tens of meters long, with their thickness varying from a fraction of centimeters to several centimeters, often showing sharp deflections of steeply dipping across foliations in the host rocks. The CGGC shear zones mostly grew in simple shear strain with kinematical vorticity number $W_k = 0.8$ -1 (Dasgupta et al., 2015).

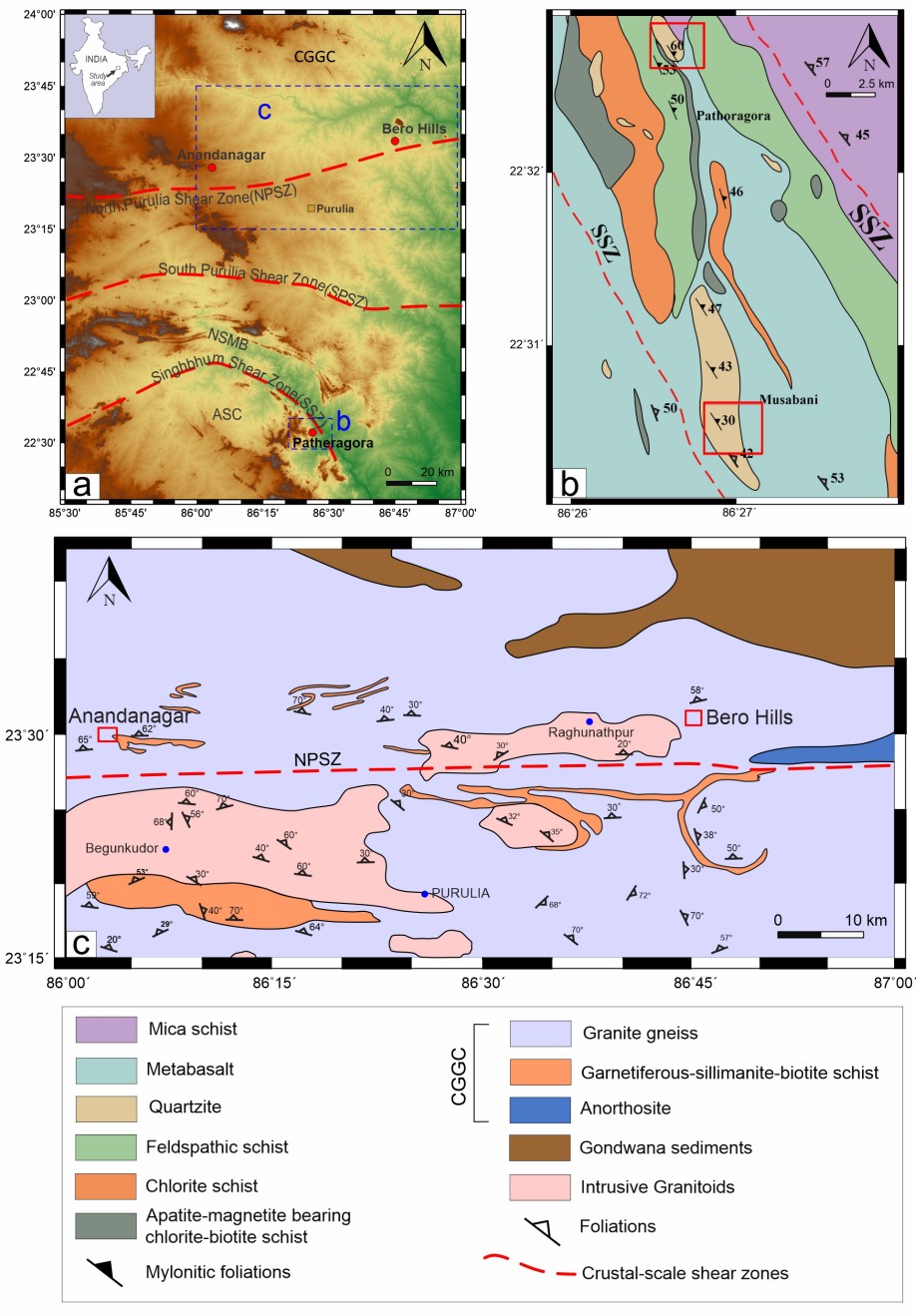

**Figure 1.** A simplified geological map of (a) the East Indian Precambrian craton, showing the dispositions of the Singhbhum Shear Zone (SSZ) and the Chotanagpur Granite Gneiss Complex (CGGC) (modified after Mukhopadhyay and Deb (1995); Mazumder et al. (2012); Roy et al. (2021, 2022)). Detailed geological maps of the two major study areas: (b) Pathoragora and Musabani region and (c) Purulia region. ASC: Archean Singhbhum Craton, NSMB: North Singhbbhum Mobile Belt, SPSZ: South Purulia shear zone, NPSZ: North Purulia shear zone. The red squares indicate the field areas.

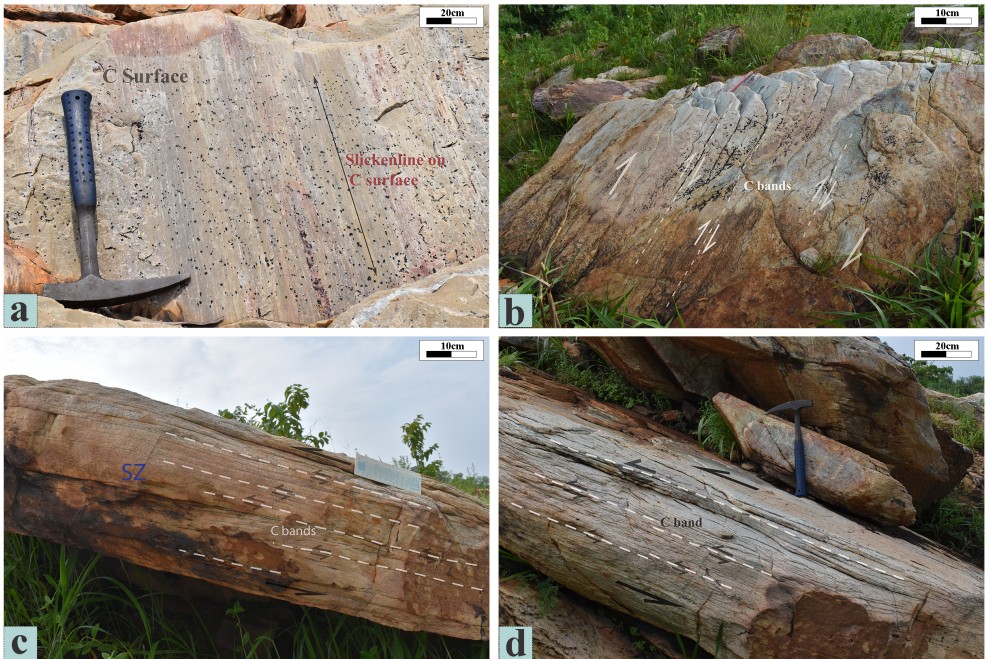

**Figure 2.** : Field examples of C dominated shear zones hosted in quartzites observed in Pathoragora and Musabani areas of SSZ. Notice strongly developed sub-parallel to parallel C-shear bands with varying spatial band density. (a) Slickenlines observed on the C surfaces indicating slip along these planes, (b-d) Intense shear localization along C bands of varying spatial densities. The bands are characterized by marked grain size reduction.SZ: Shear Zone.

## 2.2 Macro-structural characteristics of SSZ rocks

Sheared quartzites in Pathoragora show closely spaced, macro-scale NE-dipping (20° to 60°) shear surfaces (C). Their exposed counterparts profusely contain slickenlines, indicating dominantly down-dip slip motion in the shear zones (Fig. 2a). At this location the macroscopic shear structures are characterized by a single set of parallel bands (called C bands in analogy with C fabrics in mylonites), except some local gentle undulations. The C spacing varies on a wide range (2mm to 7cm). The sheared rocks are markedly devoid of S foliations on macroscale, as reported from Type II SC Mylonites by Lister and Snoke (1984). Sheared quartzites in Musabani area also show strongly developed C bands, marked by drastic grain size reduction (Fig. 2c and d). The band structures always dip in the NE direction, however, with varying magnitudes, from gentle (~20°) to steep (~45°) dips. They are laterally quite persistent, where a single C band is traceable over several meters in the down-dip as well as strike directions. The C bands are heterogeneously developed in the sheared rock, resulting in a strong variation in their spatial density ($\lambda^* = $ (width of shear zone/interband distance) =3.93 to 183.15) (Fig. 2b and c). Extremely close-spaced C-bands in places give rise to the appearance of a typical penetrative foliation, as widely reported from C-mylonites (Fig. 2d).

We measured the C spacing, normalized to the effective local shear zone thickness, as indicator of rheology, which will be discussed later (Section 2.4). Macroscopic S foliations are characteristically absent in the sheared quartzites.

Their microstructural studies under optical microscope reveal that C band domains are the regions of extreme grain refinement by dynamic recrystallization (Fig. S5). In places, the bands localize isolated slip surfaces, often occupied by minerals, such as biotite and various oxides. The matrix domains also show recrystallization microstructures, but with grain sizes markedly greater than those observed in shear bands. The recrystallized grains locally develop grain elongation in some areas, although they do not form any strong S fabrics. The microstructural characteristics of sheared quartzite are further detailed in
Supplementary (S2).

### 2.3    *Shear zones in CGGC and their internal structures*

Field investigations at Bero hills revealed sub-vertical shear zones in a granite gneiss at varied scales, with their thickness ranging from a few centimeters to more than a meter and their lengths extending up to tens of meters. Their internal structures are constituted by a combination of spaced C bands and penetrative foliations (called as S foliation, in analogy with S fabrics
in mylonites, in the foregoing description), consistently forming angular relationship between them (Fig. 2.3). Individual C bands show varying thicknesses (2.7mm to 5.1cm) and the inter-band spacing also varies on a wide range (9cm to 1.8m). The bands are typically characterized by grain size reduction, which could be detected macroscopically in the field. In places, they contain undeformed elongate pods of the host rock as remnant masses, with their long directions oriented along the bulk shear plane. The shear zones have extensively developed penetrative foliations at angles to the shear zone boundary, often forming
an anastomosing network with the C bands. This distributed S-foliation forms the lowest angle with C bands close to the band structure, which increases away from the shear band. The average angle of foliation to the principal shear direction in this location varies in the range of 15° to 30°. Some domains within a shear zone remained virtually undeformed, as reflected from the absence of C bands as well as distributed foliations. Microstructural studies of shear zone rocks show recrystallization-assisted extreme grain size reduction and secondary mineralization preferentially along shear-parallel slip surfaces in the band
domains. The inter-band regions, on the other end, consist of much larger recrystallized quartz grains, forming a well-developed shape fabrics (S) at angles to the shear direction (see Supplementary Section S2, 3 for further details). Shear band domains often contain large porphyroclasts with strong asymmetric drag structures in agreement with the overall shear sense (Fig. S7e). All these microstructural signatures indicate that the shear zones evolved through localization of high-strain zones (domains of grain refinement with slip surfaces) in a viscous deformation regime (discussed further in Supplementary S2). We evaluated the
area of macroscopic S foliation development, calculated by ratio between the domain consisting of homogeneous S foliation and the total area of the shear zone as a measure of distributed deformations in the shear zones. The calculated values range from 0.22 to 0.83, implying that the shear-accommodation mechanisms by distributed strain accumulation varies spatially in viscous shear zones.

     Most of the shear zones in Anandanagar area are hosted in porphyritic gramite gneisses (Fig. 4a-d), with their lengths ranging
from a few centimetres to more than 100 m and thickness varying from a fraction of cm to tens of centimetres. Some of them are hosted in the quartzo-feldspathic pegmatites. Their internal structures are dominated by distributed S foliations, showing little

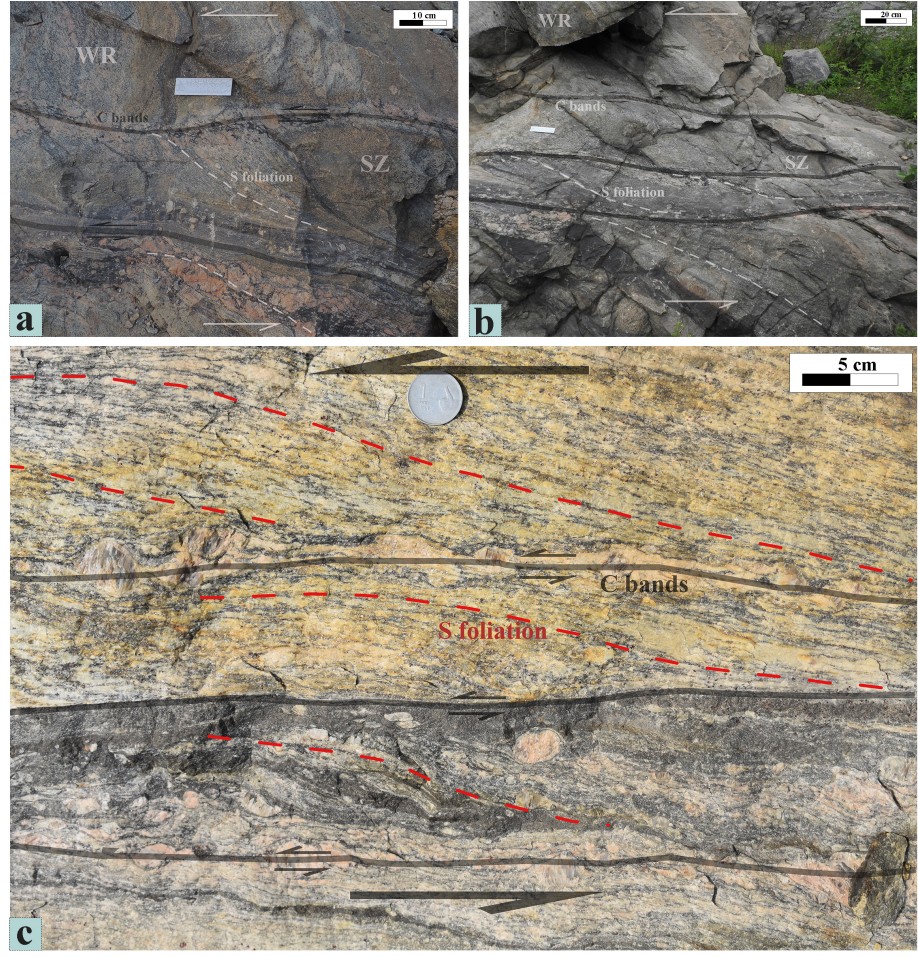

**Figure 3.** Outcrop-scale viscous shear zones containing penetrative S foliations in close association with widely spaced C bands; Bero Hills region of CGGC. (a - c) Sigmoidal patterns of S foliations in domains between C bands in the shear zone. In places, fluid assisted quartzo-feldspathic materials occur locally in C bands.

The overall foliation trends occur persistently at an angle 20° to the C band direction. WR: Wall Rock; SZ: Shear Zone.

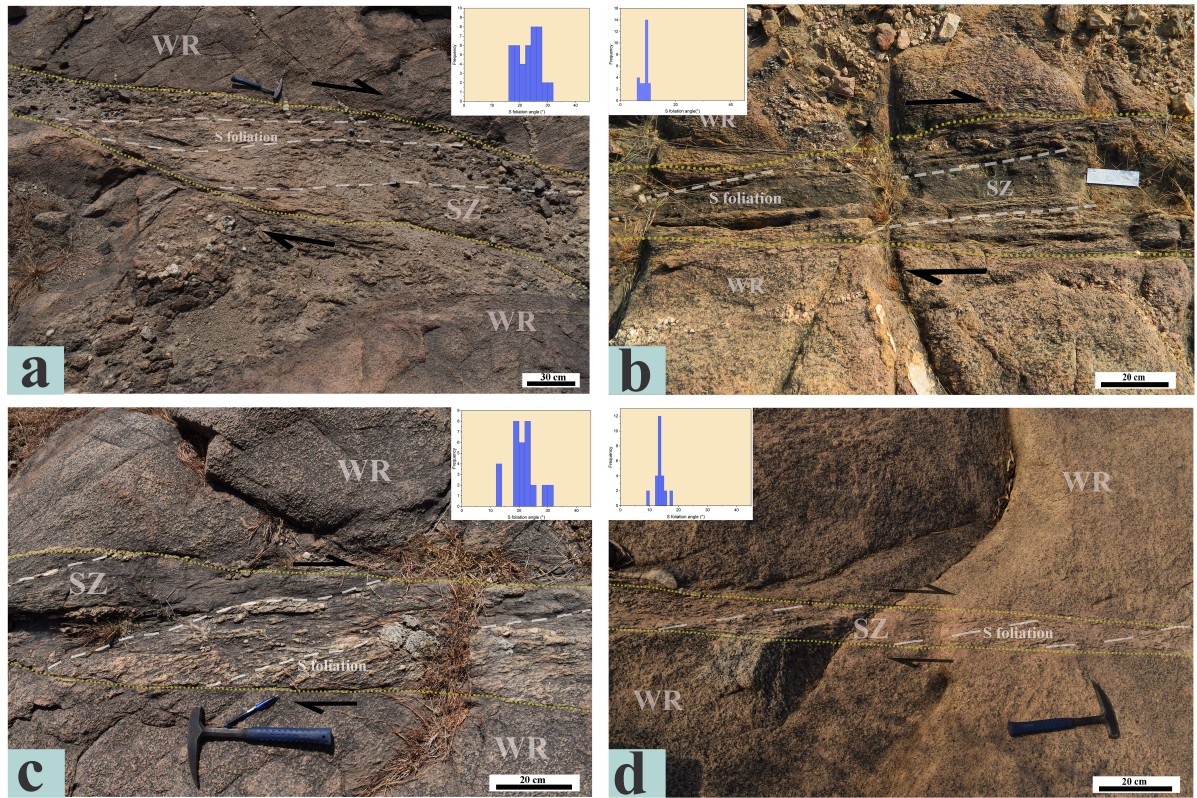

**Figure 4.** Field examples of S dominated viscous shear zones from Anandanagar area of the CGGC terrain. (a-d) Shear zones accommodating shear entirely by distributed strain accumulation, forming S foliations hosted. Notice complete absence of shear bands in them. Insets show the histogram plots of S foliation orientation ($\theta$) with respect to the shear direction. WR: Wall Rock; SZ: Shear Zone.

or no macroscopic shear-parallel bands. Under microscope these rocks are found to contain a persistent shape fabric, defined by the preferred orientations of constituting mineral grains at an angle to the shear direction (Supplementary S2, Fig. S8). The microscale shape fabrics corresponds to the macroscopic S foliation observed in the field. The major constituting minerals, quartz and feldspar have undergone extensive recrystallization, implying a dominant role of crystal-plastic mechanisms in shear deformations (discussed further in supplementary sections S2 and S3). The shear zones are generally devoid of any drag zone at their interface with the host rocks, barring a few locations where they show foliation drag and offset of across-shear zone minor veins. The shear zones in pegmatites show obliquely orientated penetrative S foliations at varying angles to their boundaries (15° to 35°). Assuming simple shear kinematics, the S-angles yield a finite shear strain of 1.6 in these shear zones. Shear zones in porphyritic granite similarly show obliquely oriented S foliations (Fig. 4d), leaving some protoliths of undeformed host rock within them. To summarize, shear zones in Anandanagar have accommodated shear dominantly by distributed strain (i.e., S foliation development), virtually with little C band-assisted deformation partitioning.

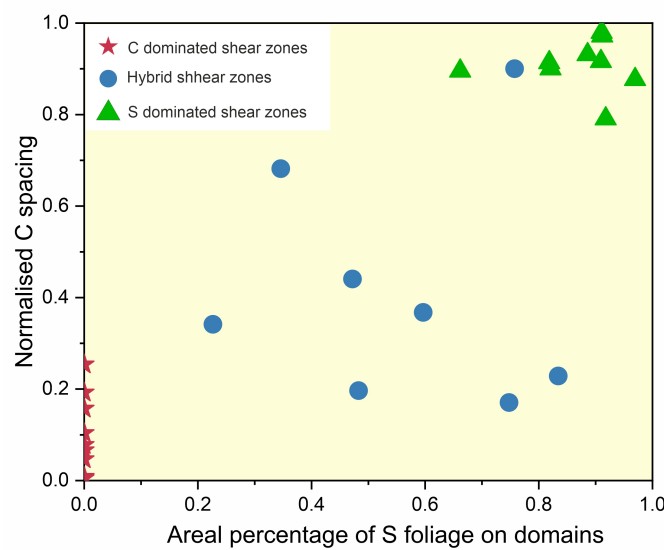

**Figure 5.** : Graphical plots of C band spacing versus areal fraction of S foliation domains. The plots delineate three distinct fields for S- and C-dominated, and hybrid shear zones. The data are collected from shear zones in the CGGC and the SSZ. The C band spacing is normalized to shear zone thickness.

### 2.4 *A synthesis of the field observations*

The relative development of distributed S foliation and localized C bands in shear zones of our study areas (SSZ and CGGC), as described in the preceding section suggests two extreme shear accommodation mechanisms (summarised in Supplementary S4). In SSZ they accommodate a large amount of shear along localized C bands, leaving the inter-band regions as domains of relatively weak development of S-foliations. In contrast, shear zones in Anandanagar areas are dominated by distributed viscous deformation, with little or no macro-scale shear band formation. Some shear zones in CGGC have evolved in a hybrid mode, where the two mechanisms: distributed viscous strain and localized C band formation operate equally. Based on the field data, we constructed a field diagram of shear zones with varying C versus S dominance using normalised C spacing and areal percentage of S foliation domains. C-dominated shear zones lie in the lower region of the field diagram while S-dominated shear zones (large areal percentage of S foliation) occupy the extreme right regime of the field diagram. The shear zone field data cluster to form distinct regions in the diagram (Fig. 5).

## 3 Numerical Modelling

### 3.1 Basic premises

Several earlier authors have used mechanical models in terms of viscoplastic rheology, where viscous properties are coupled with a yield criterion and post-yield weakening mechanisms to constitute the rheological framework (Regenauer-Lieb and

Yuen, 2003; Mancktelow, 2006; Moresi et al., 2007a). We adopt a similar rheological approach to constrain the macroscopic scale shear-accommodation mechanisms in shear zones described in the preceding section. Microstructural characteristics (details provided in supplementary S2, S3) suggest that shear zone deformations occurred dominantly by crystal-plastic creep, resulting in grain-size reduction by dynamic recrystallization. This microstructural evidence supports our consideration of viscous rheology for the shear zone modelling. Furthermore, the structural features described in the previous section reveal that shear deformations have also localized high-strain zones in the form of shear bands, which show extreme grain refinement by recrystallization, implying an abrupt increase of shear rates locally in the shear zone. They are often associated with band-parallel isolated slip surfaces on microscales. All these grain-scale features indicate that shear zones underwent differential mechanical weakening, which we implement by combining a pressure-dependent yield criterion with the viscous rheology. Within this rheological premise, the shear zones are modelled as narrow zones of viscous-plastic materials, undergoing Stokes flow, as applicable for incompressible slow, non-inertial viscous fluid flows (Gerya and Yuen, 2007; Jacquey and Cacace, 2020; Ranalli, 1997). The shear zone materials are assumed to yield plastically at threshold stresses. The model shear zones are approximated to crustal rheological regimes, setting their geometrical characteristics (e.g., length to thickness ratios) and kinematic conditions applicable to the corresponding natural prototypes.

### 3.2   Mathematical formulation

In our shear zone modelling, the mathematical formulation considers a two-layer system that embodies naturally formed shear zones, where a mechanically weak zone is hosted within a relatively stronger surrounding rock (wall) (Mancktelow, 2006; Pennacchioni and Mancktelow, 2018; Cawood and Platt, 2021). This modeling approach is effective for developing numerical, time-evolving, and dynamically consistent shear zone models in 2-D Cartesian domains within the theoretical framework of computational fluid dynamics (CFD). The CFD simulations, in this instance, assume an incompressible Boussinesq fluid flow, approximating the long-time (million years) scale kinematic state of Earth's lithospheric deformations. We employ the following continuity and momentum conservation equations in the CFD modelling:

$$\nabla . u_i = 0, \tag{1}$$

$$-\nabla P + \nabla \cdot \tau_{ij} + \rho g_i \; = \; 0 \tag{2}$$

where $u$ denotes velocity, $P = 0.5(\sigma_{xx} + \sigma_{yy})$ is the pressure, $\tau$ is the deviatoric stress tensor, and $\mathbf{g}$ denotes gravitational acceleration. In Eq. (2), the inertial forces are neglected, as applicable to long-term flows in Earth's interior. The deviatoric stress tensor ($\tau_{ij}$) is derived by subtracting the isotropic part from the total stress tensor ($\sigma_{ij}$). Assuming incompressible viscoplastic rheology, the deviatoric stress tensor ($\tau_{ij}$) can be equated with the strain rate tensor as :

$$\tau_{ij} = 2\eta_{\text{eff}}\dot{\varepsilon}_{ij} = \eta_{\text{eff}} \left( \frac{\partial u_i}{\partial x_j} + \frac{\partial u_j}{\partial x_i} \right) \tag{3}$$

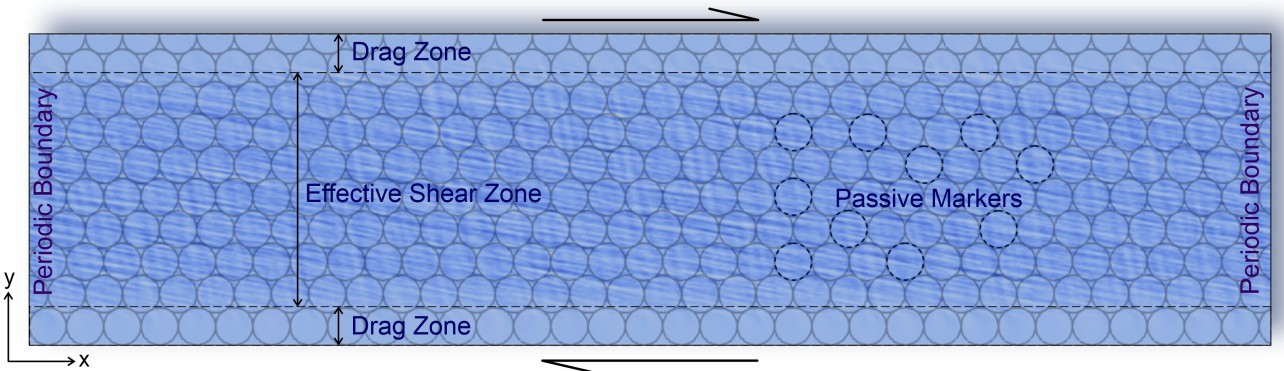

**Figure 6.** Representative initial shear zone model used for numerical simulation experiments run with a visco-plastic rheological approximation. The model considers a three-layered mechanical structure: a core, flanked by drag zones, hosted between two undeformable boundaries as commonly observed in geological settings. The model domain is imprinted with initially circular passive markers to determine the finite strain distributions across the shear zone. Further details of the model boundary conditions are provided in the text.

where $\dot{\varepsilon}_{ij}$ is the strain-rate tensor and $\eta_{\text{eff}}$ is the effective viscosity, which includes viscosities $\eta_v$ (viscous creep) and $\eta_p$ (plastic creep) in their reciprocal form (Sandiford and Moresi, 2019) given as

$$\eta_{\text{eff}} = \left( \frac{1}{\eta_v} + \frac{1}{\eta_p} \right)^{-1} \tag{4}$$

The shear zone modelling is implemented in an incompressible visco-plastic rheological framework (Ranalli, 1995), conventionally represented by a series connected frictional block – dashpot mechanical model. This consideration allows us to resolve the deviatoric strain-rate into two parts: viscous ($\dot{\varepsilon}_{ij}^v$), and plastic ($\dot{\varepsilon}_{ij}^p$) in the form

$$\dot{\varepsilon}_{ij} = \dot{\varepsilon}_{ij}^v + \dot{\varepsilon}_{ij}^p \tag{5}$$

where,

$$\dot{\varepsilon}_{ij}^v = \frac{1}{2} \frac{\tau_{ij}}{\eta_v} \qquad\qquad \dot{\varepsilon}_{ij}^p = \begin{cases} 0 & \text{if } J_2 < \sigma_{yield} \\ \chi \frac{\tau_{ij}}{2J_2} & \text{if } J_2 \geq \sigma_{yield} \end{cases} \tag{6}$$

$J_2 = 3(\sigma_{jj})^2 - \frac{1}{2}\sigma_{ij}\sigma_{ij}$ represents the second stress invariant, that determines the plastic creep at the yield point and $\chi$ is a plastic multiplier.

Natural shear zones accommodate shear mostly by large long-term permanent strains. Thus, the elastic (reversible) strain component is negligibly small ($\dot{\varepsilon}_{ij}^e \to 0$), as compared to the permanent viscous and plastic strains. Therefore, we ignore its effects on the shear zone models. Post-yield viscous weakening of the material, where the modified viscosity decreases non-linearly with increasing plastic strain, is introduced to incorporate strain-softening rheology in the shear zone models. Plastic yielding is implemented by locally rescaling the effective viscosity in such a way that the stress does not exceed the yield

stress, also known as viscosity rescaling method (Willett, 1992; Kachanov, 2004; Glerum et al., 2018). The effective viscosity is then given by

$$\eta_{\text{eff}} = \frac{\tau_{ij}}{2|\dot{\varepsilon}|} \tag{7}$$

where $\dot{\varepsilon}$ represents the second invariant of $\dot{\varepsilon}_{ij}$. $\eta_{\text{eff}}$ signifies the reduced viscosity once the yield limit is attained. It is to be noted that the nonlinearity introduced by the post yield plastic rheology is solved from eqn 7 iteratively. This equation provides the necessary conditions to introduce weakening in the shear zones with parametrically imposed strain rate weakening laws, as implemented in many earlier viscoplastic models (Mancktelow, 2006; Glerum et al., 2018; Roy et al., 2021). In such yield-controlled rheology, the effective viscosity decreases non-linearly with increasing finite strain (Fig. S10). To model shear zones formed at middle to lower crustal depths (Duretz et al., 2014; Gueydan et al., 2014; Reber et al., 2015), as applicable to our field studies, we have considered a linear relation of the yield stress ($\sigma_{yield}$) with pressure ($P = -\frac{1}{2}\sigma_{ij}$), and modelled the yield behaviour by employing a pressure-dependent plasticity (Drucker-Prager) criterion (Roy et al., 2021; Rast and Ruh, 2021). Based on this criterion, a yield function, $F$, can be defined in the following form

$$F = \sigma_{yield} - \sqrt{3}\sin(\phi)P - \sqrt{3}C(\gamma_{\text{pl}})\cos(\phi) \tag{8}$$

where $C(\gamma_{\text{pl}})$ is the material cohesion, expressed as a function of plastic strain $\gamma_{\text{pl}}$, and $\phi$ is the angle of internal friction. The cohesion is assumed to weaken with increasing accumulated plastic strain as,

$$C = C_i + (C_f - C_i)\min\left(1, \frac{\gamma_{\text{pl}}}{\gamma_o}\right) \tag{9}$$

where $C_i$ is the initial cohesion and $C_f$ is the final cohesion of the shear zone material. $\gamma_{\text{pl}} = \int_0^t \dot{\varepsilon}_p \, dt$ indicates accumulated plastic strain in regions where the yield limit is reached, and $\gamma_o = 0.1$ is taken as the reference strain. No syn-deformational healing of the cohesion is implemented in the present models.

### 3.3 Model Setup

Based on the mathematical formulation described in the preceding section, we developed 2D shear zone models using the open-source code Underworld 2 (http://www.underworldcode.org/) to solve the mass and momentum conservation equations (Eqs. 1 & 2) under incompressible conditions to obtain the pressure and velocity conditions within the shear zone domain. This code works within a continuum mechanics approximation, and has been extensively used to deal with a range of geological and geophysical problems (Beall et al., 2019; Mansour et al., 2020; Roy et al., 2024). As explained in Moresi et al. (2007b) and Mansour et al. (2020), the code discretizes the geometrical domain into a standard Eulerian finite-element mesh and the domain is coupled with the particle-in-cell approach (Evans and Harlow). To implement the particle-in-cell approach, the code discretizes the material domain into sets of Lagrangian material points, which allow us to find material properties that are history-dependent (in the present case, the plastic history of material) and can be tracked over the entire simulation run. Physical properties of the shear zone materials, such as density and viscosity are mapped by particle indexing. Our modelling excludes the effects of temperature diffusion and any inertia in the system.

The shear zone models were developed in a $4L \times L$ rectangular domain, where $L$ represents the reference length scale. The model domain, occupied by incompressible visco-plastic materials, is discretized into quadrilateral mesh elements comprising of $584 \times 324$ elements. We considered a three-layer model architecture (Roy et al., 2022) to simulate viscous shear zones, consisting of an intensely sheared core, flanked by drag zones on its either side, hosted in unsheared rocks, which structurally resemble those observed in the field (Fig. 2-4). The procedure to model the three-layered structure is discussed in detail in

the Supplementary section S6. We imposed the following velocity boundary conditions in the shear zone models. The side model boundaries were subjected to a periodic boundary condition (Fig. 6), whereas the bottom and the top boundaries were assigned a prescribed velocity in the horizontal direction, keeping the overall strain rate constant throughout the simulation. The boundary velocities produced a dextral simple shear movement, where the maximum tensile stress ($\sigma_1$) axis is oriented at an angle of $45°$ to the bulk shear direction.

Our model simulations were run by varying the three major parameters: 1) initial viscosity ($\eta_v$), 2) initial cohesion ($C_i$), and 3) bulk shear rates ($\dot{\gamma}_b$) in the system, where the first two characterize the rheology and the third represents the bulk kinematics of a shear zone. We vary $\eta_v$ between $1\eta_o$ and $100\eta_o$, and similarly, $\dot{\gamma}_b$ between $1\dot{\gamma}_o$ and $100\dot{\gamma}_o$, where $\eta_o$ and $\dot{\gamma}_o$ are the background viscosity and shear rate, respectively. All material parameters used in the simulations are summarized in Table 1.

**Table 1.** Numerical model parameters and their values

| Parameters | Symbol | Natural Values | Numerical Input Values |
|:---:|:---:|:---:|:---:|
| Model length | $L$ | $4\ km$ | 4 |
| Model width | $W$ | $1\ km$ | 1 |
| Model reference strain rate | $\dot{\gamma}_o$ | $2.7 \times 10^{-14}\ s^{-1}$ | 1 |
| Model reference density | $\rho g$ | $27000\ kgm^{-2}s^{-1}$ | 1 |
| Model reference viscosity | $\eta_0$ | $1 \times 10^{21}\ Pas$ | 1 |
| Initial Cohesion | $C_i$ | $27\ MPa$ | 1 |
| Angle of friction | $\phi$ | $25° - 30°$ | $25° - 30°$ |
| Maximum Yield stress | $\sigma_{\max}$ | $1000\ MPa$ | 37 |
| Minimum Yield stress | $\sigma_{\min}$ | $10\ MPa$ | 0.37 |

### 3.4   Model shear zone characteristics

The simulations presented in this study primarily aim to constrain the rheological and kinematic conditions determining the shear accommodation mechanisms, leading to development of C, S, and C-S structures of crustal shear zones described from our field observations (Fig. 2-4). This section presents three sets of simulations to demonstrate the distinctive modes of strain evolution in model shear zones, designated as reference model (RM).

In the first set of RM1 simulations, run with $\eta_v = 1\eta_0$, $\dot{\gamma}_b = 0.5\dot{\gamma}_0$, and $C_i = 2C_0$, model shear zones accommodate the

applied shear entirely by uniformly distributed continuous viscous deformations (Supplementary Video S1), revealed from deformed elliptical shapes of initially circular passive markers in the model (Fig. 7a-ii). The homogeneous strain continues to

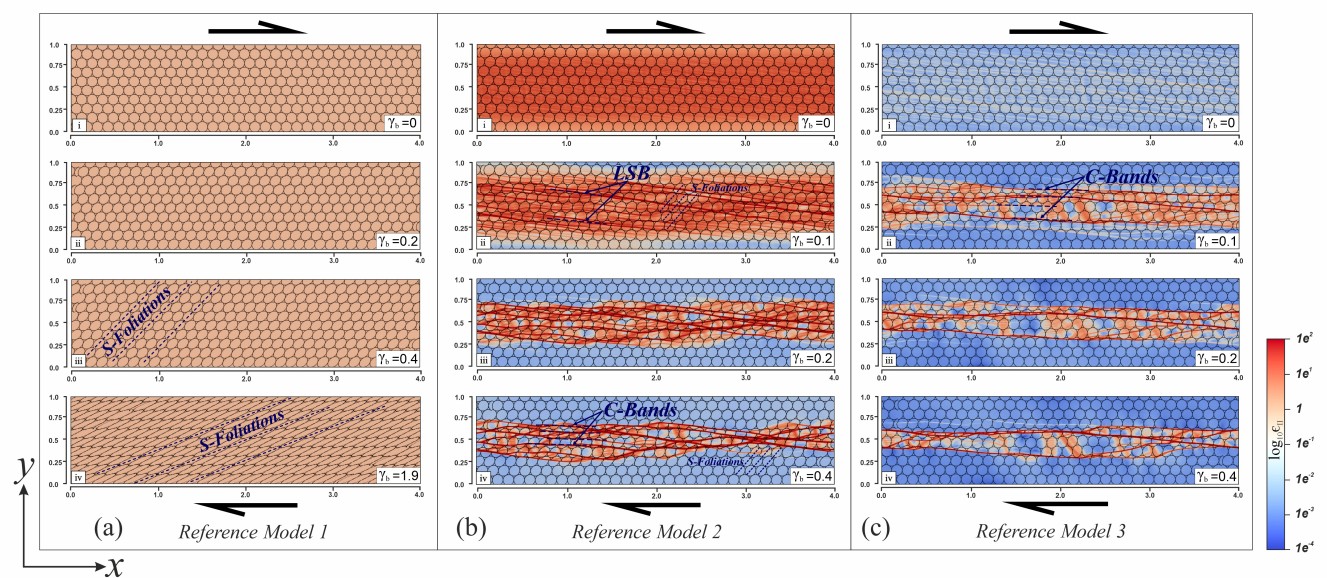

**Figure 7.** Evolution of shear zones in the three reference models under dextral shear: (a) Reference model 1: $\eta_v = \eta_0$, $\dot{\gamma}_b = 0.5\dot{\gamma}_o$, and $C_i = 2C_0$ (b) Reference model 2: $\eta_v = 50\eta_0$, $\dot{\gamma}_b = 10\dot{\gamma}_o$, and $C_i = 1C_0$ (c) Reference model 3: $\eta_v = 100\eta_0$, $\dot{\gamma}_b = 3\dot{\gamma}_o$, and $C_i = 1C_0$, where $\eta_0$ and $\dot{\gamma}_o$ represent the background viscosity and shear rate, respectively. The color bar represents the logarithmic magnitude of strain rate invariant. Notice a transition of shear accommodation mechanism from homogeneously distributed strain accumulation to spaced shear band localization from RM 1 to 3. RM2 produces low-angle shear bands (LSB) in the initial stage of shear deformation ($\gamma_b < 0.2$), subsequently replaced by shear parallel C bands with progressive shearing movement.

increase with progressive shear, but without showing any tendency to localize discernible shear bands throughout the simulation runtime (Fig. 7a-iv). Increasing finite bulk shear ($\gamma_b$) results in flattening of passive ellipses with decreasing inclinations of their major axes to the bulk shear direction in agreement with that obtained from strain ellipses for homogeneous simple shear equation (Fig. 8):

$$\tan 2\theta = \frac{2}{\gamma_b} \tag{10}$$

This finding suggests that this type of model shear zones accommodates bulk shear entirely by homogeneous viscous strain, allowing no strain localization associated with pressure-dependent yield. This primarily occurs strain as the flow stress condition always lies below the yield point, which is evident from the clear absence of shear bands (Fig. 7a-iv). The RM1 simulations, consistently fail to produce shear band structures at any stage of shear zone evolution. Consequently, homogeneously distributed deformation emerges as the principal mechanism for accommodating shear.

The second reference model (RM2) simulations were run at high bulk shear rates ($\dot{\gamma}_b = 10\dot{\gamma}_o$) and high viscosity ($\eta_v = 50\eta_o$), while keeping $C_i = 1C_0$. At $\gamma_b = 0.09$, the RM2 simulation run developed finely spaced low-angle ($\theta \approx 13°$) shear bands and a few sporadic high-angle ($\theta \approx 85°$) shear bands (Supplementary Video S2). The densely packed low-angle bands impart a

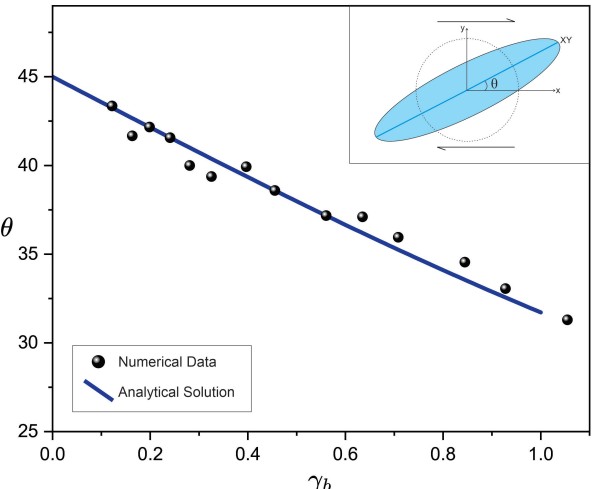

**Figure 8.** Inclinations ($\theta$) of the major axis of finite strain ellipse (shown in inset) as a function of the finite shear ($\gamma_b$), obtained from RM1 numerical simulations (black dots). Passive circular markers were taken in the model domain to obtain the strain ellipses. It is noteworthy that the numerical data regression agrees well with the analytical solution (solid blue line, Eq.10), implying that the model shear zone has accommodated shear by homogeneously distributed strain accumulation.

low-angle foliation in the shear zone (Fig.7b-ii). Increase in $\gamma_b$ gives rise to penetrative distributed viscous strains, revealed from elliptical shapes of the initially circular passive markers in the model, and the distributed strain accumulates steadily (Fig. 7b-ii) until $\gamma_b$ reaches a threshold value (0.15) when the band structure underwent a drastic transformation with the appearance of shear-parallel C bands at $\gamma_b> 0.15$ (Fig.7b-iii). The shear zone ultimately accommodates shear by a set of sub-parallel, wide-spaced ($\lambda^* =\sim 0.263$) C surfaces, forming a weak network, although locally with the earlier formed low-angle

bands (Fig.7b-iv). P-bands localize sporadically, with a tendency to network with the principal C-bands. During the post-yield period the C-band assisted shearing becomes the dominant shear accommodation mechanism, manifested in disruption of the deformed passive markers (Fig. 7b-iv). The model shear zones thus accommodate the bulk shear initially by uniformly distributed viscous strain, switching to localized C band-assisted shear accommodation.

The third reference model (RM3) simulation is assigned an extremely high initial viscosity ($\eta_v = 100\eta_o$) and a moderate

bulk shear rate ($\dot{\gamma}_b = 3\dot{\gamma}_o$), produces a band growth pattern remarkably different from those observed in RM2 simulations (Supplementary Video S3). At $\gamma_b = 0.08$, the model first develops a set of low-angle shear bands at an angle of $\sim 15°$ to the shear direction, characterized by their narrow, long, and closely spaced ($\lambda \approx 0.006$) geometry, along with sporadically occurring thick high-angle shear bands ($\lambda \approx 0.19$). With progressively increasing bulk shear ($\gamma_b > 0.1$), shear band formation becomes the primary shear-accommodation mechanism, leading to a complete structural transformation into thick, widely

spaced ($\lambda^* \approx 0.26$) shear-parallel C bands with virtually no traces of low-angle shear bands (Fig. 7c-ii). RM3 simulations

show shear-zone evolution with little or no distributed viscous deformations, as indicated by the undeformed shapes of the initially circular markers in the model (Fig. 7c-iii). The post-yielding slip in the C-bands resulted in intense local deformations along the trace of these bands (Fig. 7c-iv). However, low-angle shear bands that formed at a low bulk shear ($\gamma_b < 0.1$) had little effect on the deformation of passive markers, implying that their slip was negligible.

## 3.5 C-band Localization versus Distributed Deformations: Rheological Constraints

Our simulation results reveal a functional relationship between the two distinct competing shear-accommodation mechanisms (distributed strain versus localized shearing) with the bulk shear rate ($\dot{\gamma}_b$) and the initial viscosity ($\eta_v$) of shear zones. The plotted data illustrates these relationships in a log-scale representation of $\dot{\gamma}^*(=\frac{\dot{\gamma}_b}{\dot{\gamma}_0})$ vs $\eta^*(=\frac{\eta_v}{\eta_0})$ space for a given value of $C_i = 1C_0$, where $\dot{\gamma}_0$ and $\eta_0$ are the background shear rate and viscosity, respectively, representing the regional shear rate and wall rock viscosity (see Fig. 9). The field diagram indicates that low $\dot{\gamma}_b$ and $\eta_v$ values facilitate homogeneously distributed strain accumulation in the entire shear zones. However, this mechanism is completely replaced by shear localization in the form of C-bands with increasing $\eta_v$. The field diagram also shows that increasing $\dot{\gamma}_b$ initially results in homogeneous strain-assisted shear accommodation, but once the yield point is surpassed, it switches to the C-band mechanism. It is noteworthy that the individual fields for each of these strain accumulation processes can change depending on the involvement of more complex grain-scale processes in shear deformations.

## 4 Discussion

### 4.1 Shear accommodation mechanisms and their crustal conditions

Field observations presented in Section 2 reveal contrasting structural characteristics in crustal shear zones in SSZ and CGGC. For example, sheared rocks in Anandanagar shear zones contain S foliations oblique to the shear zone boundaries with little or no macroscopic C bands, whereas those in Patherogora and Musabani extensively display shear parallel C bands, but no macroscopic S foliations. On the other hand, shear zones in Bero hills contain both the structural features (summarised in Supplementary Section S4). Modelling of viscous shear zones in the framework of incompressible viscous rheology with a pressure sensitive yield criterion suggests that these varying internal structural characteristics originate from their contrasting shear-accommodation mechanisms. S dominated shear zones accommodate their bulk shear by spatially distributed viscous deformations across the whole shear zone, whereas those dominated by C bands accommodate shear by strain localization, forming spaced shear bands.

These two end-member mechanisms of crustal shear zones are also manifested in their internal structural architectures. Shear zones are architecturally more homogeneous, showing uniformly developed S foliation in case of the first mechanism and geometrical continuity of across-shear zone passive markers, e.g., quartz veins. In contrast, the second mechanism results in heterogeneous structural architectures, characterized by a close association of localized zones of intense shearing, resulting in grain refinement and fluid assisted mineralisation along these zones, leaving adjoining regions relatively undeformed.

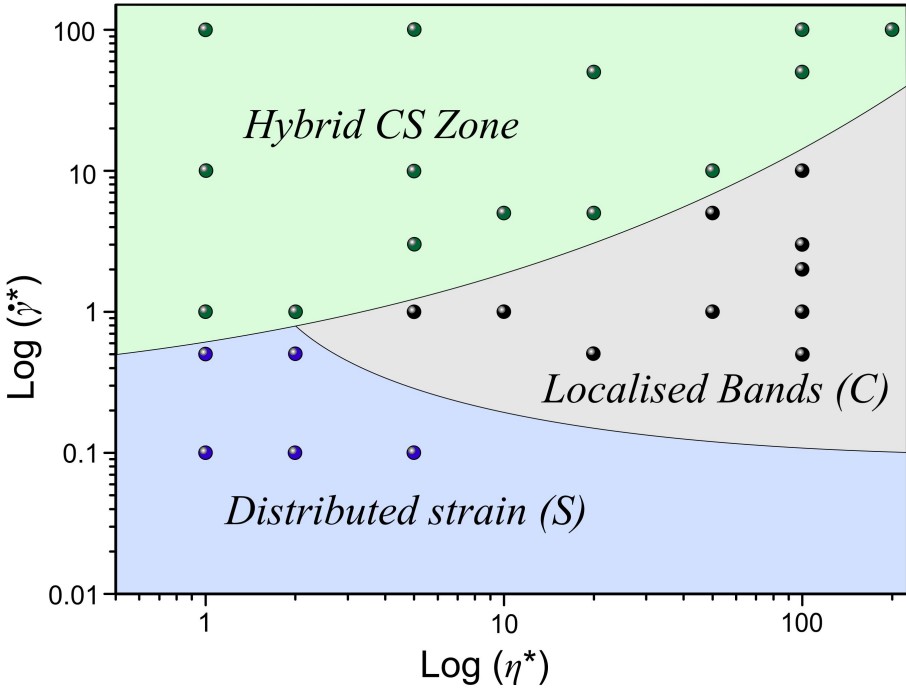

**Figure 9.** Field diagram of the three shear-accommodation mechanisms in a space defined by log-scale representation of $\gamma^*$ ($\frac{\dot{\gamma}_b}{\dot{\gamma}_0}$) and $\eta^*$ ($\frac{\eta_v}{\eta_0}$), constructed based on numerical simulations with $C_i = 1C_0$. $\dot{\gamma}_0$ and $\eta_0$ represent the background shear rate and viscosity in the regional setting. It is noteworthy that increasing initial viscosity of crustal shear zone materials facilitates the C-band assisted shear-accommodation mechanism.

Earlier authors (Mancktelow, 2006; Meyer et al., 2017) have shown that a transition from homogeneously distributed shear accommodation mechanism to its localizing counterpart can be explained by a drop in fluid pressure within the shear zones (pressure dependent yield), together with enhancement of viscous mechanisms by additional effects of fluid assisted mineralog-
390   ical changes and grain-size reduction due to recrystallization. The microstructural analyses of shear zone rocks from our field area reveal that they are characterized by dynamic recrystallization and grain refinement. There are also evidences of fluid migration and mineralization along local slip surfaces, which probably further enhanced the weakening process to localize strain along these zones. Such softening processes can be equated with the strain-rate dependent weakening factor implemented in our numerical models.
395   The numerical simulations indicate that the type of internal strain accommodation mechanism is sensitive to the bulk strain rate in the tectonic setting. Crustal deformations at low strain rates favour development of internally homogeneous viscous shear zones by the first mechanism (Fig. 7a), as observed in a region within the CGGC terrain that record regional deformations. Such internally homogeneous shear zones have also been extensively reported from different geological terrains across the globe, e.g. Ramsay et al. (1983) Fossen and Cavalcante (2017) Pennacchioni and Mancktelow (2018). On the other hand, high strain-

rate kinematic conditions that generally occurs in specific crustal regimes, e.g. terrain boundaries and accretionary wedges in subduction zones, can give rise to internally heterogeneous shear zones, characterized by strong strain partitioning in numerical simulations (Fig. 7c). Our model interpretations are consistent with the field observations from the Singhbhum Shear Zone (SSZ), at the boundary between the Proterozoic NSMB and the Archean Singhbhum Craton where the strain rates are expected to be high (Ghosh and Sengupta, 1987). Several workers have shown evidences of slip and vein emplacement along shear surfaces in viscous shear zones from subduction margins (eg. Platt et al. (2018); Ujiie et al. (2018); Tulley et al. (2022). They often contain undeformed lenses formed by networking of anastomosing shear band structures (Carreras et al., 2010). These heterogeneities are generally attributed to fluid mediated mineralogical transformations along subducting plates or inherent lithological heterogeneities within the plate boundaries. Based on our numerical simulations, we propose that shear zones in initially mechanically homogeneous systems at high strain rates can evolve to become structurally heterogeneous due to the domination of shear-accommodation mechanisms by localized shearing, followed by various syn-shearing transformations, e.g., fluid-assisted mineral transformations.

Relative viscosity ( $\eta^*$ ) and initial cohesion ( $C_i$) of shear zone rocks also play a pivotal role in determining the internal architectural characteristics in our model shear zones. Low $\eta^*$, as observed in high-temperature environments at deep-crustal levels, favour a homogeneous structural characteristics of our model shear zones (Fig. 7a), which becomes extremely heterogeneous as $\eta^*$ is increased to a threshold values ( $\eta^* = 2$). In contrast, reducing cohesion, which may be assisted by syn-shearing micro-scale fracturing and fluid migration (Wu and Lavier, 2016; Menegon et al., 2021), can facilitate shear band formation, resulting in heterogeneous structures within the shear zones (Fig. 7b ).

## 4.2 Mylonite characteristics: indicator of shear accommodation mechanisms

Mylonites are the typical representative rocks of viscous shear zones, produced by intense shearing and grain-size reduction, accompanying characteristic structural fabrics formation. Among them, C and S are most common planar fabrics in mylonites, as discussed in Section 1. However, field observations show wide variations in the relative development of these two structural manifestation in the macroscopic scale, ranging from S-foliations to C band dominated structures in shear zones. This type of macroscopic variation can be compared with those reported from mylonites based on microscale fabrics, which are classified as Type I (mylonites containing S and C fabrics) and II (C fabrics) (Lister and Snoke, 1984). Although a broad spectrum of Type I and type II SC mylonites have been reported from natural viscous shear zones (Gates and Glover III, 1989; Mukhopadhyay and Deb, 1995; Cacciari et al., 2024), their origin deserves a discussion, especially in the context of shear localization mechanics. Correlating the shear zone fabrics with the model strain fields, it appears that shear deformation localize at significantly higher strain rates preferentially in shear parallel narrow discrete planar zones resulting in intense grain refinement by dynamic recrystallization to form C-fabrics. This process can similarly occur in the macroscopic scales to produce C band as obtained from our numerical models (Fig. 7c). On the other hand, distributed homogeneous viscous strain in the entire shear zone can flatten primary mineral grains to reorient them along XY planes of the finite strain ellipsoid, forming S-fabrics at an angle to the shear direction. The fabric development syncronously participates in dynamic recrystallization to produce a fraction of smaller grains. These mechanisms can give rise to macroscopic S foliation in the distributed viscous domains if the shear

zone rocks were initially coarse grained or rich in platy minerals such as muscovite, biotite and chlorite, as observed in the CGGC (Fig. 4). Our simulation results reveal that shear zones growing in high-viscosity rocks localize densely packed shear surfaces (Fig. 9) with little or no distributed homogeneous strain. This mechanical condition can thus produce shear parallel foliation without penetrative fabrics, as observed in ideal Type II mylonites. Lowering the initial viscosity transforms the shear accommodation mechanism to facilitate the distributed strain accumulation setting (Fig. 9) a favourable condition for the development of selectively S fabrics, i.e., Type I mylonites. To summarize, the shear accommodation mechanism by shear band formation under high-viscosity conditions and/or high shear rate (Fig. 9) favours C-dominated mylonites (Type II), whereas that by distributed strain development under low-viscosity conditions with high strain rate (Fig. 9) facilitate Type I SC-mylonites in crustal shear zones. It is to be noted that the mylonitic fabric developments, as discussed above, can be much more complex due to various grain-scale factors, such as crystallographic orientations, polymineralic assemblage and grain boundary processes, e.g., grain boundary migration (Finch et al., 2020, 2022). The fabric growth in such cases generally track the local strain fields developed under the influence of micro-scale heterogeneities (Mukhopadhyay et al., 2023).

A prolonged debate, which is still lively centres on the synchronous versus sequential development of S and C foliations in the evolution of shear zones. From field evidences Berthé et al. (1979) showed the coexistence of S and C at shear zone boundaries, both with their increasing spatial density towards the shear zone core. Their observation goes in favour of synchronous foliation development. However, this interpretation is not universally accepted and confronted with an alternative proposition, claiming that S foliations precede C localization. This structural sequence can occur when shear zones first accommodate shear by distributed strain, followed by localized yielding to localize C surfaces. Lister and Snoke (1984) elaborated categorize mylonites into three types based on the temporal relationship between S and C fabrics. They showed that in some cases, both S and C foliations form synchronously in the same shear event, whereas in some other settings, e.g. an older metamorphic complex, they grow in two successive events, where S foliations formed in the earlier event are overprinted by C localization during the later deformation event. Their study finds a third possibility for a complex structural sequence, where transient flow patterns during ongoing shearing causes C foliations to align in the shortening field, leading to their folding and formation of a new set of S fabrics. Numerical simulations by Finch et al. (2020) and analogue experiments conducted by Dell'angelo and Tullis (1989) on quartzites provide additional insights. At low strain levels, the dominant foliations are primarily S foliations, with C' shear bands to form preferentially in the core regions of the shear zones, however, without distinct slip in highly strained samples. Studies have also shown that there can be a considerable time gap between the formation of S-fabrics and C bands (Bukovska et al., 2013; Bukovská et al., 2016) On the other hand, experiments conducted on quartz-feldspar aggregates produce weak S-C foliations, implying that monomineralic rocks, such as quartzite may not readily form S-C foliations. Burlini and Bruhn (2005) suggested distributed viscous strain and shear localization as the two competing processes. According to them, a brittle event occurs prior to onset of yielding, otherwise the shear zone would develop distributed strain over the entire sample. Based on our numerical model results, we propose distributed viscous strain development and localized shearing as two end-member shear accommodation mechanisms, which can occur synchronously, although one dominating over the other depending on the rheological conditions of the shear zones. Consequently, a shear zone can continue to accommodate viscous strain even after the C band formation or vice-versa in progressive shear.

## 4.3 Rheological controls and their tectonic implications

Earlier field and experimental investigations as well as numerical simulations have dealt with the problem of strain partitioning in polymineralic rocks, and demonstrate the distinctive roles of distributed viscous and localized strains in shear zones (Mancktelow, 2006; Katz et al., 2006; Burlini and Bruhn, 2005; Misra et al., 2009; Finch et al., 2020; Tokle et al., 2023). The present study takes into account the pivotal effects of bulk rheology and kinematics on strain partitioning, and shows that the interplay of bulk viscosity, shear rate, and cohesion of materials within shear zones as the primary determining factors of

shear zone processes. Depending on these factors, as discussed in the preceding section, shear zones undergo shearing entire by viscous strains with little or no internal shear localization that could produce porosity bands (Katz et al., 2006) to enhance the permeability in shear zones. In such situations they would hardly act as pathways for fluid or melt migration. In contrast, same shear zones can profusely produce shear bands under a favourable condition of the aforesaid factors that will provide effective permeability for fluid migrations, as often recorded in the form of shear-parallel veins and pegmatites (Creus et al.,

2023; Koizumi et al., 2023). The role of crustal shear zones in deep-earth fluid transport processes, as shown by many authors (Cox, 2002; Fusseis et al., 2009; Spruzeniece and Piazolo, 2015; Précigout et al., 2017) thus depends primarily on the mode of shear zone evolution controlled by distributed strain versus localized shear accommodation mechanisms.

Several recent studies (Vissers et al., 2020; Allison and Dunham, 2021; Lavier et al., 2021; Mildon et al., 2022) have reported a number of phenomena, such as earthquake generation and frictional melting (pseudotachylites) from shear zones that indicate

the occurrence of extremely high shear rates, which are difficult to explain from relatively slow shear kinematics. Evidently, there must be some weakening mechanisms in shear zone processes to amplify the shear rates by many orders, e.g., $10^{-12}$ $s^{-1}$ to $10^{-3}$ $s^{-1}$ (Kelemen and Hirth, 2004). Understanding the process of such shear rate enhancement is critically important to interpret many important deep-earth processes in viscous regimes, such as earthquakes within the subducting lithosphere, especially at depths ranging from 50 to 300 km. It is important to note that viscous mechanisms primarily govern deformation

at high pressures and temperatures corresponding to such depths, and thus cannot account for seismic fault events (Platt et al., 2018; Gou et al., 2019). To address this problem, a range of explanations have been hypothesized for the mechanisms of intermediate-depth earthquakes, such as dehydration embrittlement and thermal runaway. Dehydration embrittlement, as proposed by Hacker et al. (2003) and Frohlich (2006), result from enhanced pore pressures, resulting from fluids released from metamorphic reactions. On the other hand, thermal runaway occurs due to interplay between weakening triggered by

various factors and the temperature-dependent rock rheology (Kelemen and Hirth, 2007; Andersen et al., 2008; John et al., 2009; Braeck and Podladchikov, 2007; Thielmann and Kaus, 2012). Field evidence supporting thermal runaway includes the presence of pseudotachylites (e.g., Andersen et al. (2008)); in fact, some of them are reported from viscous settings far below the brittle-viscous transition. Our study provides an alternative explanation for strain localization and strain-induced slip within shear zones. Model results indicate that shear zones localize shear bands with significantly reduced effective viscosity.

This strain softening phenomena are generally attributed to factors, such as grain size reduction, mineral reactions, and the development of crystallographic preferred orientation. The model shear bands locally enhance strain rates by ∼2-3 order. The viscoplastic model employed in this study incorporates a pressure-dependent yield criterion, contributing to strain localization

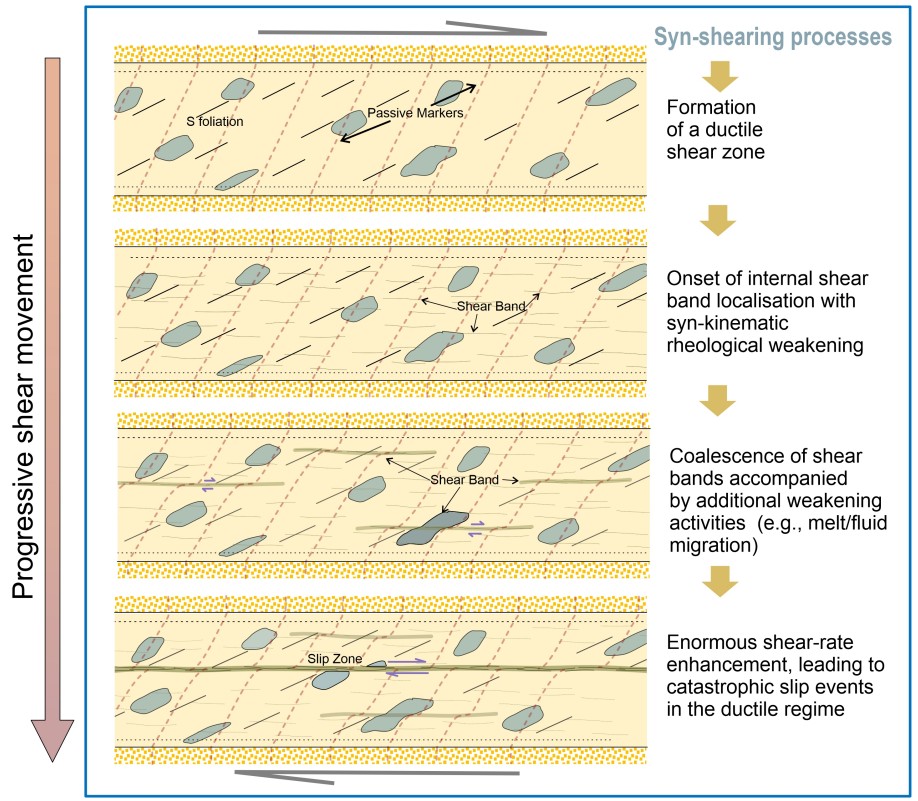

**Figure 10.** A visual representation illustrating various developmental phases of crustal-scale shear zones subjected to dextral shear motion. The gradual escalation of shear, coupled with diverse syn-kinematic strain softening mechanisms, may culminate in catastrophic slip events within the viscous regime.

along zones of strain softening(Fig. 10). We ran a set of simulation by increasing the degree of strain softening in this criterion to test how much softening is required to attain higher order strain rates. The simulation results suggest that exceptionally large shear rates can localize along shear bands in the viscous regime due to the effects of additional shear softening processes e.g., porosity driven metamorphic fluids/melts. For example, theoretical calculations predict dramatic reduction of effective viscosity by an order of $\sim$2-3 (Holtzman, 2016), on addition of a small fraction ($\sim 5\%$) of melts/hydrothermal fluids. We thus propose that viscous shear zones can produce earthquakes (Fig. 10) if their evolution is modulated by shear band-assisted strain-rate enhancement during a yield event, accompanying additional syn-shearing weakening agencies such as fluids and melts to further enhance the strain rates. The mechanical setting under such high strain rates can undergo a viscous to brittle rheological transition, which switches a sudden release of the stored elastic strain energy in the form of seismic waves (Schubnel et al., 2013; Prieto et al., 2017).

## 4.4  Model Limitations

The rheological (visco-plastic) approach adopted in the present modelling explains the shear accommodation mechanisms in crustal shear zones on a macroscopic level. There is a limitation in this approach as it does not account for fabric developments under the influence of complex grain scale phenomena, such as strain perturbations due to crystallographic or mineralogical heterogeneities. The numerical modelling also excludes transient thermal effects of syn-shearing processes, like shear heating. Such heating can play an additional role to facilitate the mechanical weakening and the process of shear localization (Kelemen and Hirth, 2004). Considering that there is no large lithological variations in the present study area, the field diagram shows the modes of shear accommodation as a function of shear zone viscosity and strain rate for a constant initial cohesion ($C_o$). However, $C_o$ can vary to a large extent in geological terrains due to various reasons, such as lithology, and this variation can influence the field of each shear accommodation mechanism shown in the diagram (Fig. 9). In addition, the individual fields can vary depending on the rheological complexity, which is not explored in the present model. Despite these limitations, this study provides a first-hand insight into the mechanics of strain partitioning in crustal shear zones.

## 5  Conclusions

This study combines field observations and numerical simulations to show that shear-parallel strain localization and homogeneous viscous deformation as two competing mechanisms of deformation accommodation in crustal shear zones. We can summarise the findings along the following points.

a) Shear zones produce strongly varying internal structural characteristics, modulated by the two competing shear-accommodation mechanisms. The localized shearing mechanism gives rise to shear-parallel band formation whereas distributed strain accumulations result in penetrative S foliation development tracking the XY plane of the finite strain ellipsoids.

b) These two competing mechanisms are controlled by the following three factors: bulk strain rate, bulk viscosity and initial cohesion. Increasing bulk viscosity and strain rate or reducing cohesive strength transform the shear accommodation mechanisms from distributed homogeneous strain accumulation to localized C band formation.

c) For a specific threshold of shear rate ($\dot{\gamma}^*$=0.5) under given viscosity and cohesion, viscous shear zones can produce both S foliations and C bands (e.g., SC mylonites) by distributed strain accumulation and localized shear-accommodation mechanisms, respectively.

d) Homogeneous versus heterogeneous internal architectural characteristics of shear zones hosted in an initially homogeneous rocks depend largely on the shear accommodation mechanisms. The distributed strain accumulation mechanism gives rise to more uniform structural architectures in low-strain ($\dot{\gamma}^* < 0.5$) crustal conditions, as compared to those produced by the localized shearing mechanism in high-strain ($\dot{\gamma}^* > 0.5$) crustal environments.

*Code availability.* The authors confirm that all the data used to support the findings of this study are available within the manuscript and as Supporting Information. All aspects of UNDERWORLD 2 (Beucher et al., 2022) can be downloaded and checked in this link: https://doi.org/10.5281/zenodo.6820562

.

*Data availability.* The relevant data supporting the conclusions are present in this manuscript, the Supporting Information and in the repository (https://doi.org/10.6084/m9.figshare.25563030.v2) (Chatterjee et al., 2024)

.

*Author contributions.* PC conducted the field investigations, performed the analyses, and prepared the initial draft. AR conducted the numerical simulations and contributed to the initial draft writing. NM conceptualized the central research ideas and overall planning, supervised the methodologies, and revised the initial draft.

*Competing interests.* The authors declare that there are no conflict of interests.

*Acknowledgements.* We thank two anonymous reviewers and the Editor Federico Rossetti for their incisive comments and constructive suggestions. PC acknowledges DST-SERB funded project (JBR/2022/000003) for providing doctoral research fellowship. AR gratefully acknowledges CSIR, India for awarding research fellowship grants (09/096(0940)/2018- EMR-I). The DST-SERB is acknowledged for supporting this work through the J.C. Bose fellowship (JBR/2022/000003) to NM.

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
