# Peer review of "Localized shear versus distributed strain accumulation as shear-accommodation mechanisms in crustal shear zones: Constraining their dictating factors"

_EGUsphere, 2024_

## Author Comment (AC1)

**Reply to comments of Reviewer 1**

**Comment:** The manuscript describes modeling results and field examples of shear bands and shear zones. The results are presented in a clear, well-written, and concise text. The modeling results appear reasonable and sound. The field examples are presented at the mesoscale observation level. However, the manuscript requires some major revisions before it can be published. The first problem that I see lies in some confusion of terms that is related to nomenclature:

**Response:** It is nice that the reviewer has appreciated the presentation of our model results, considering them as "reasonable and sound". We thank the reviewer for providing us with insightful comments and suggestions for revising the manuscript, which have been carefully addressed in this revised version, as explained below.

**Comment:** For a large part of the text, especially the modelling part, the authors use a continuum mechanics rheology nomenclature, consistent with their modelling approach, which is continuum-mechanics-based. In this nomenclature "plastic" refers to "pressure-sensitive, temperature-insensitive" deformation with a yield criterion, and "viscous" refers to "temperature-sensitive and pressure-insensitive" deformation without a yield criterion. These definitions are not clear to all geologists or may be used differently by them and therefore should be defined in the introductory section.

**Response:** Thanks for this relevant discussion by the reviewer on our rheological considerations and the use of rheological nomenclatures: 'viscous' and 'plastic'. In the Introduction section (Ln: 132-136) we discuss the visco-plastic rheology of shear zones considered in the present modelling, where the two terms: viscous and plastic rheology are defined, as suggested by the reviewer. We also elaborate the basis of this rheological consideration in the modelling section (Ln: 226-236).

**Comment:** Furthermore, the term "ductile" is problematic in geology and rock mechanics. "Ductile" in rock mechanics primarily refers to brittle, distributed deformation, e.g., cataclastic flow, and in this sense, the brittle-ductile-transition is a purely confining-pressure-dependent transition from discrete fractures to zones of distributed cracking. Friction-controlled sliding may agree with the term "plastic" in the purely rheological sense defined above. However, the term "ductile shear zone" is used by most geologists as a zone where viscous deformation processes (intracrystalline plasticity or diffusion creep) dominantly accommodate the strain and thus a viscous rheology prevails. Obviously, from the short outline above, it becomes clear that the terms "ductile" and "plastic" have very different meanings in the different communities. Large parts of the discussion suffer from this confusion of terms. Again, the terms should be clearly defined and probably the terms "ductile" and "plastic" (without the prefix "crystal") should be avoided or their use should be checked for consistency in every instance.

**Response:** We agree with the reviewer that the terms: 'ductile' and 'plastic' are used with different meanings in different communities. We used the term- 'ductile' to mean distributed viscous deformations without macroscopic fracturing in shear zones. In fact, shear zones of our present study show viscous deformations, which is evident from extensive dynamic recrystallization on grain scales, i.e., signatures of crystal-plastic creep mechanisms, as rightly pointed out by the reviewer. The present version provides a detailed account of viscous rheological consideration in this version (Main text- Ln: 226-236, Supplementary section S3). To avoid confusion, we have now replaced the term- '*ductile*' with '*viscous*' in the entire manuscript, as suggested by the reviewer.

In the Introductory section we briefly discuss the principal rheological approaches, for example, power-law viscous and visco-plastic rheology used in previous studies for shear band localization (Ln: 84-88).

In our modelling approach we apply a yield criterion to initiate strain localization (shear band formation) within a visco-plastic macroscopic rheological framework. The plastic yield criterion is chosen based on the occurrence of localized high-strain zones, which show extreme grain size reduction and shear-parallel slip surfaces (c-surfaces) on microscales, as indicated by the reviewer. These features are described in detail in Supplementary section (S2) of the revised version. Based on the reviewer's suggestion, we now drop the term- plastic in the entire manuscript to avoid confusion, except in the expression of macroscopic model rheology (i.e., visco-plastic rheology). These issues are elaborately discussed in the Modelling method section (Ln: 226-236) and Supplementary section (S3).

**Comment:** The second problem of the manuscript lies in the lack of microstructural analysis in the field examples. The microstructures could provide information on the deformation mechanisms in each shear band or -zone. Once the deformation mechanism is established, rheological consequences are implied. E.g., for cataclastic-frictional microstructures (perhaps the quartzite examples?), the rheology may be "ideal-plastic" in the rheological sense or "ductile" in the rock mechanics sense, but not in the common structural geology sense. The S-foliation-dominated microstructures may indicate crystal plastic or diffusion-creep-type deformation mechanisms and therefore could imply dominantly "viscous" deformation in the rheological sense. The discussion would become much clearer, far more relevant, and less speculative with such information provided. Furthermore, rate-dependent and viscosity-related inferences are made from the mechanical modeling and discussed. Such a discussion should only use the field examples when deformation mechanisms are established for the examples – otherwise the field examples are black-box cases.

**Response:** We greatly appreciate this insightful comment and discussion by the reviewer, indicating the need of a microstructural support to justify the choice of rheology in the shear zone modelling. This revised version provides a new section to

describe deformation-associated microstructures of shear zone rocks. As rightly pointed by the reviewer, we find extensive grain size reduction by dynamic recrystallization (crystal-plastic mechanism). This allows us to consider an overall viscous rheology of the shear zones, which is clarified in this version (Ln: 226-236, see also Supplementary S3). Microstructural studies also reveal sharp variations in recrystallized grain sizes, delineating zones of strain-rate enhancement due to commencement of yielding in shear zone rocks. These high-strain zones often contain grain scale shear-parallel slip surfaces, often filled with secondary minerals, such as biotite, chlorite and oxides. Based on these microstructural characteristics, we consider a yield criterion to develop macroscale strain localization in a viscously deforming system, as described in a visco-plastic rheological model. The revised version includes a set of microstructural analyses to support our rheological considerations for shear zone modelling. These new additions are, however, placed in the supplementary section (S2) to maintain the manuscript length. We sincerely thank the reviewer for this excellent suggestion.

Detailed comments:

**Comment:** Line 21: omit "intense"

**Response:** Corrected, as suggested. (Ln: 21)

**Comment:** Lines 83-107: The discussion should include the possibility that the S- and C- fabric elements may not develop simultaneously as poposed by Berthe et al. 1979. Recent studies by Bukovska et al. 2013, 2016 indicate a different origin and should be mentioned and discussed here.

**Response:** We thank the reviewer for this important suggestion. We have now included these studies in both the Introduction and Discussion sections of the updated version (Ln: 109-112 and 460-461).

**Comment:** Line 86-87: there is important experimental evidence for the formation of shear bands in the semi-brittle deformation regime, and this should be considered here, too: Pec et al. 2016, Marti et al. 2017, 2018, 2020, Schmocker et al. 2003.

**Response:** The reviewer has rightly noted that shear bands can form in a semi-brittle deformation regime. A brief statement on this point is included in the Introduction. We have cited the relevant works provided by the reviewer. Thanks for this suggestion (Ln: 81-84).

**Comment:** Line 95: definition of terms "viscous" and "plastic", see introductory comments above. It will not be clear to most geologists how or why the terms viscous and plastic are used in a distinguishing of differing sense here. Furthermore, it is not clear why the strain accommodating processes in S and C bands have to be different.

**Response:** Based on the introductory comment by the reviewer, we have used the term: 'viscous' in place of 'ductile'. The basis of viscous rheological consideration has also been explained (Ln: 132-136 and 226-236). We used the term: 'plastic' only in the expression of visco-plastic rheology, where a yield criterion is coupled, which is explained in the preceding response.

It is to be noted that we use the nomenclature: C-Bands to mean macroscale bands of strain localization parallel to the shear direction, surrounded by regions of distributed viscous deformations, where shear bands are absent on macroscopic scales.

In our model strain localization results from a yielding process, involving weakening and reduction in the effective viscosity (Eq. 7) that in turn enhances the shear rates. C-bands accommodate strain at much higher shear rates, as reflected from dramatic decrease in recrystallized grain sizes. In addition, shear-parallel slip surfaces are found to localize more preferentially in them. These aspects are described in the revised version (Ln: 226-237) and Supplementary section (S3).

We appreciate this nice point noted by the reviewer.

**Comment:** Line 99: "accommodates" instead of "accommodate"

**Response:** This is corrected. Thanks. (Ln: 114)

**Comment:** Line 100: how is it determined that the deformation in the localized zones is not viscous?

**Response:** The text is modified to clarify the reviewer's comment. We describe C bands as localized zones of high strain, where viscous deformation can occur, as rightly commented by the reviewer. In fact, the revised version shows microstructural evidence (extreme grain size reduction by crystal-plastic creep and recrystallization), implying that a shear band also takes part in intense viscous deformations. This context is discussed in different parts of this version (Ln: 170-175 and 226-236). Thanks for raising this important point.

**Comment:** Fig. 1: please give scales in km, not just in degrees of latitude and longitude. CGGC does not appear in the maps but in the text – please indicate the abbreviation in the maps or refer to other units (NPSZ?)

**Response:** Corrected. The scales are provided in km, as suggested by the reviewer. Thanks.

**Comment:** Lines 158-162: by foliation you refer to a S-foliation? Please specify.

**Response:** Yes, it refers to a S-foliation, which is now specified (Ln: 185).

**Comment:** Fig. 3: the C-bands show a coarse grain and have a melt-like appearance within the feldspar-biotite matrix. Such melt segregations will have a different mechanical property compared to the matrix. Please comment on this aspect, especially with respect to the relevance to modeling and in terms of rheological development.

**Response:** In places, the shear zones had emplacement of quartzo-feldspathic materials along C-bands. This feature is now mentioned in the revised figure caption. The  bands localized under the influence of the pre-emplacement rheological condition in the shear zone, and they later controlled the emplacement process in the course of shearing. However, the associated fluid activities might have acted as rheological weakening factor to facilitate the process of shear localization, as considered in our numerical modelling formulation (Eq. 7). This issue is briefly addressed in the discussion (Ln: 387-394).

**Comment:** Fig. 4: the shear zones are considerably coarser grained than what is termed "wall-rock" here and appear to have a melt-origin, while the wall rock does not show clear evidence for melt. Again, as in Fig. 3, a considerably weaker rheology is expected for these shear domains. Modeling such structures appears difficult: have the melt segregations formed first, so that they localize the deformation? In such a case, a homogeneous matrix cannot be assumed for modeling. Or has melt material filled pre-existing shear bands? If this is the case, why are such bands so dilatant?

**Response**: This shear zone has actually developed in a pegmatitic body, which initially consisted of very large crystals that underwent size reduction during shear deformation. However, their grain size still remained coarser than the gneissic host rock. We have replaced this panel with another example to maintain consistency in the presentation.

**Comment:** Lines 178-180: C-band formation appears to be in contrast with viscous deformation here – why? Please define or describe the difference between viscous deformation and localized shear band formation. Why should localized deformation not be viscous? Commonly, shear bands can be considered localized zones of viscous deformation.

**Response:** Yes, the reviewer has rightly suggested that shear bands can also undergo intense viscous deformations, as revealed from grain size contrasts. In addition, shear bands contain sporadic microscale slip surfaces, which are generally absent in the domain of distributed viscous deformations. The updated version presents detailed description of microstructural evidence in supplementary section (S2) to show the distributed and localized viscous deformation. This issue is also clarified in the main text (Ln: 170-175, 188-194 and 202-206).

**Comment:** Line 276: "accommodates" instead of "accommodate"

**Response:** Corrected. Thanks. (Ln: 332)

**Comment:** Lines 276-278: this statement implies that plastic yield will produce some strain localization. In principle, plastic deformation may produce homogeneous strain – depending, in part, on the definition of "plastic". That is why it is important to define the terms, see introductory comments

**Response**: The term 'plastic' was used to model the yielding phenomenon in the shear zone materials, accompanying synkinematic weakening and reduction of effective viscosity, as explained in the preceding response. This sentence is modified to avoid confusion in our expression (Ln: 333).

**Comment:** Lines 318-322: the terms viscous and plastic appear to be used in a strictly continuum mechanics rheological sense here. As many geologists may have a somewhat different understanding of these terms, it is important to explain them in the introductory part. Furthermore, the difference in plastic and viscous strain accommodation mechanisms may follow from the modeling, but the mechanisms are not demonstrated for the field examples. For a complete discussion, this aspect of the analysis needs to be performed or at least some evidence for supporting an interpretation of different deformation processes needs to be given.

**Response:** The reviewer has correctly noted that the present model uses the terms: viscous and plastic, in a rheological sense within a framework of continuum

mechanics. This point is now clearly stated in this version (Ln: 132-136). Based on the reviewer's suggestion, we now include microstructural descriptions (main text, Ln: 170-175, 188-194 and 202-206; supplementary section S2) to show the rheological basis of shear zone deformations, as addressed in our preceding responses.

**Comment:** Lines 323-329: these few lines discuss very important aspects of definitions and identification of deformation mechanisms in conjunction with rheology. The identification of viscous deformation mechanisms is fairly straightforward from thin sections. As for "plastic" deformation in the rheological sense, this can manifest itself in cataclastic deformation processes, because these are pressure-sensitive. Such processes can also be identified from thin sections. The term "ductile" in some rock mechanics literature (e.g. Byerlee) can include distributed brittle deformation (e.g. cataclasis). Geological literature often refers to ductile as a viscous deformation. See general introductory remarks above.

**Response:** We greatly appreciate the suggestion for providing grain scale characteristics as a support for rheological considerations. The revised version includes microstructural description of C-bands and the bulk regions, i.e., outside the bands. They allow us to account for viscous rheology with a plastic yield criterion in our shear zone model (Ln: 226-236). The microstructural descriptions are placed in the Supplementary section (S2) to maintain the main manuscript length. Many thanks for this constructive suggestion.

**Comment:** Lines 330-344: the occurrence of different types of shear zones is less dependent on the tectonic setting but, instead, strictly temperature- and strain rate-dependent. Of course, higher temperatures and lower strain rate favor viscous deformation, whereas cataclastic deformation processes dominate in lower temperature regimes and at higher strain rate.

**Response:** The reviewer has correctly pointed out that the shear zone type is less dependent on the tectonic settings. We actually meant that the tectonic setting can largely control strain rates, which in turn determine the type of shear zone. This part has been modified to clarify our expression (Ln: 295-417). We appreciate this discussion by the reviewer.

**Comment:** Line 351: "ductile shear zones" – see general comments above. Probably, this term should be avoided altogether.

**Response:** We have dropped the term- ductile in this version of the manuscript.

---

## Author Comment (AC2)

**Response to Reviewer # 2's comments**

**Comment:** This article analyzes homogeneous distributed strain versus shear band formation in shear zones in Eastern India and tries to find governing rheological and kinematic parameters controlling the formation of natural C, S, and CS fabrics using 2D, visco-plastic numerical shear deformation models. The paper is well written and clearly structured. The description and explanation of the natural fabrics in the different outcrops is detailed and accurate and the discussion tries to connect the numerical results and the natural observations in a good way.

However, I do struggle with some of the terminology, especially in combination with the numerical models. In the current way and how the authors argue for a correlation between the numerical results and the natural fabrics, the numerical model setup seems inappropriate to describe the natural fabrics. Especially, considering the simple rheology of the numerical models and no discussion on how a simple visco-plastic rheology can explain or represent/mimic complex microscopic features leading to strain localization and shear band formation.

**Response:** Thanks for appreciating the main theme of this study, and also raising important issues on some terminology used in the numerical modelling and correlation between the numerical results and the natural fabrics. We have adopted a continuum mechanics approach based on a visco-plastic rheological approximation, with a principal objective to investigate the problem of distributed versus localized (i.e. band formation) deformations in shear zones, as observed in macroscopic scales. This approach treats the continuum as an idealized medium that represents an average rheological property of granular rocks. The model thus does not replicate grain scale deformation features, as rightly pointed out by the reviewer. Evidently, granular entities are required to develop microscale fabrics in the strain fields. Several previous studies have proposed models to show the development of such fabrics in mylonites, which we briefly discuss in the revised version (Ln: 442-445). We clearly state that the present model deals with the problem of strain partitioning in shear zones from a macroscopic perspective. This issue is addressed in the Limitation section (Ln: 514-517).

We like to add that this study correlates the model results with macroscopic foliations observed in shear zones of our study areas, where the distributed deformation regions display dominantly penetrative foliations at an angle to shear direction and C-bands parallel to the shear zone boundary (called as S foliation and C-band in analogy with S and C fabrics of mylonites commonly observed in thin sections). This is explicitly indicated in the beginning of the field description (Ln: 159-160 and 179-189).

**Comment:** I like the combination of the numerical models with clear and precise natural examples. However, in the current way it is hard to believe, that a simple visco-plastic rheology is enough to explain complex natural C,S, and CS fabrics. It is

very nice to see how they connect the different fabrics with the main rheological and kinematic parameters (shear rate and viscosity). However, no discussion or explanation is made on how one could correlate shear bands due to plastic failure in the models with natural shear bands forming C fabrics, besides their simple pattern.

**Response:** As mentioned in the preceding response, the modelling approach aims to deal with the problem of macroscopic scale shear-parallel band formation, separated by regions of distributed penetrative foliations, as observed in the field. The model results enable us to define two kinematic domains: shear-parallel high-strain localization (C-band) and distributed strain fields. Evidently, the two macroscopic domains do not directly account for microscale fabric development (defined by preferred orientations of mineral grains) in viscous shear zones.

In the distributed strain fields, S foliations can grow macroscopically by tracking the XY planes of the local strain ellipsoids, assuming that the materials are capable of producing such foliations either by flattening (e.g., crystal-plastic mechanisms) of grains or reorientation of platy minerals(Ln: 430-435). On the other hand, the macro-scale shear partitioning (i.e., C-banding) that we show from our continuum model can occur also in microscales to form C-fabrics. However, the fabric developments can be much more complex due to various grain scale attributes, as shown by several previous authors. We discuss these issues in this revised version (Ln: 442-445; Lines ---)). Many thanks for this excellent comment.

**Comment:** Also, I am not convinced that the numerical results are specifically new. Figure 9 simply states, that for a certain yield stress one obtains yielding above a threshold viscosity for a given shear rate. This is expected though, since the stress increases with increasing viscosity in simple shear deformation assuming a constant shear rate! Using a different cohesion results in different regimes again!

**Response:** Thanks for this comment. It is to be noted that the main objective of this study concerns how contrasting strain localization patterns can originate from the process of strain partitioning in viscously deforming shear zones. To the best of our knowledge, the field diagram presented here is a new addition. Figure 9 shows the three types of macroscopic strain accumulation as a function of strain rate and viscosity in shear zones. We, however, do not claim the viscous deformation versus yielding phenomena discussed here as our new contribution.

Yes, cohesion would be another factor to influence the different fields shown in this figure, as rightly pointed out by the reviewer. We have varied the cohesion in our numerical models to show the differences in shear accommodation mechanisms. Considering that there is no dramatic change in lithological settings of our study areas we chose to present the field diagram as a function of dynamic/kinematic parameters (viscosity and strain rate), keeping the initial cohesion constant. However, the

cohesion can be an important parameter to influence the field of each shear accommodation mechanism shown in the figure. We have explicitly stated this issue as a limitation in the revised version (Ln: 519-522).

**Comment:** The connection between natural examples and numerical models is interesting as well as the classification of the different fabrics in the shear-rate – viscosity regime would be interesting for the geoscientific community. Thus, I believe this article, is worth publishing, however, only after a major revision. I do hope the comments below help the authors to do so.

**Response:** Thanks to the reviewer for considering this article as worth publishing and suggesting important suggestions for improvement of our study. We have included all these constructive suggestions in revising the manuscript.

Major comments

*Visco-plastic rheology and correlation with natural fabrics*

**Comment:** That the parameter combination of ductile viscosity, shear rate and cohesion result in regimes of homogeneous distributed strain, localized plastic shear bands, and a combination thereof is clear. How those patterns can be link to the natural fabrics not!

**Response:** This comment has been addressed in the preceding response. The model presented here is developed in the perspective of a homogeneous continuum to show the strain fields. Evidently, the model cannot physically produce a fabric, which requires grain-scale attributes. Several studies have simulated fabric developments in grain-scale models (cited in this version, Ln: 442-444). We assume that the continuum is capable of developing a fabric tracking the strain field. A discussion has been included in (Discussion Section, Ln: 430-435) to show the possible link of the model strain partitioning (macroscopic) patterns to C- and S- fabrics of mylonites observed in thin sections (Main text, Ln: 427-435; Supplementary section S3), as suggested by the reviewer. Thanks for giving us this nice suggestion.

In addition, we discuss how the contrasting magnitudes of accumulated strains in shear bands can produce stronger grain refinement, as compared to that in the surrounding regions (Field section, Ln: 170-175, 188-194). The revised version includes microstructural descriptions of differential grain refinement. We also document development of microscale s-fabrics defined by preferred orientations of platy minerals, like mica in shear zones dominated by homogeneously distributed strain accumulation (Field Section, Ln: 202-206). These are also presented in detail in Supplementary section S2.

**Comment:** My major concern is the rheological configuration of the numerical models and how the results of the numerical models are correlated to the natural

fabrics in the shear zones. Besides the fact, that the rheology is not purely viscous (plasticity is included), I am missing an explanation on how the numerical results can explain the natural fabrics. How does a purely viscous simple shear deformation result in S fabric (homogeneously distributed strain sure, but this does not tell us anything about the fabrics)? If this is the case, then we should see fabrics in all viscously deforming rocks. Is this really the case? A more detailed discussion or explanation thereof would be very helpful.

**Response:** We greatly appreciate this excellent discussion on the correlation between rheological consideration and natural fabrics. In this version we have clearly noted that the rheological modelling aims to address the two macroscopically observed deformation domains: localized band vs distributed strain fields. A new section has been added to provide microstructural features of shear zones in support of rheological considerations in the numerical modelling (Ln: 226-236).

The reviewer has correctly indicated that the present model does not cover the processes involved in the development of microscale fabrics. However, we have discussed the possible correlation of our macroscale model observations with the mylonitic grain scale features commonly observed under microscope (Ln: 427-435). In addition, from microstructural studies of shear zones rocks from our field areas, we have shown microstructures associated with the two macroscopic deformation domains (Ln: 170-175, 188-194 and 202-206). However, most of these discussions are placed in the Supplementary section (S2) to keep the main manuscript length.

The reviewer has indicated that the development of fabrics in microscale can be more complex due to grain scale heterogeneities, which again cause strain localization to control fabric developments, as reported from field and experimental observations. We briefly discuss this issue, and cite relevant studies in the Discussion section (Ln: 242-245).

**Comment:** This is even more concerning regarding the correlation between plastic failure and the localized shear bands. Yes, the pattern look similar and one could explain the natural fabrics via such a pattern in the numerical models. But, clearly the natural localized shear bands are not formed via plastic failure. An explanation or discussion should be given on what the plastic failure represents or mimics such that it can be linked to the natural shear zone. This is currently missing, or not very clearly described. Current research on plastic strain-weakening processes help to link plastic strain-softening and hardening to micro-physical mechanisms like a grain-size sensitive composite olivine rheology. While healing is not specifically applied in this research here, a similar way to argue for the connection between a visco-plastic rheology including strain-dependent weakening mechanisms and complex micro-physical mechanisms would be helpful. Considering the fact, that strain-localization is not only driven by grain-size reduction other potential mechanisms could be

discussed a little bit in more detail to link the numerical patterns observed in the models with the natural fabrics.

I believe a discussion in that direction would benefit the manuscript, maybe also additional models with a more complex rheology (or an argument why to stick to a simple visco-plastic rheology) is really necessary.

**Response:** Thanks for this insightful discussion, raising an important issue of plastic failure versus localized shear band formation and its link to the natural shear zone. In visco-plastic modelling of shear zones we consider a yield criterion to initiate the process of strain localization in the viscously deforming shear zones. The yielding process is coupled to a mechanical weakening factor in the mathematical formulation, which facilitates the band growth in progressive shearing. We have provided a detailed microstructural description of macroscale shear bands from field areas (Ln: 170-175, 188-194 and 202-206). The bands show extensive grain size reduction by dynamic recrystallization, which can linked to syn-kinematic mechanical weakening during the band growth. Secondly, they also show the occurrence of microscale slip surface, which can be correlated with the process of plastic yielding described on macroscale. As correctly pointed out by the reviewer, other agencies can facilitate this weakening process. We report the presence of fluid-assisted signatures along C-bands, which can be an additional weakening factor. We discuss these issues in the Discussion (Ln: 387-394).

**Comment:** Multiple micro-physical mechanisms can lead to strain localization, which are also described in the natural examples (grain size refinement in the localized shear bands), however, which dominates is still unclear. To mimic those mechanism by a simple visco-plastic rheology would be to easy, without a clear definition why, which, I believe is currently missing. Considering a more complex rheology in the discussion of the numerical results might also put the final conclusion in Figure 9 into a different perspective, since I strongly doubt that it is so simple to classify the natural fabrics by purely visco-plastic rheology.

**Response:** In the revised version we have included a new section on the microstructural features of shear zones rocks (Supplementary S2), which are discussed to support our visco-plastic rheological consideration. For example, the signatures of extensive recrystallization suggests that the shear zone grew in a viscous deformation regime. Similarly, the occurrence of abrupt reduction in recrystallized grain sizes indicate localized strain-rate enhancement. These zones also contain isolated slip surface. These microstructural evidences indicate the process of plastic yielding in the viscous deformation regime. This is the discussed in the Modelling section (Ln: 226-236).

We would like to emphasize once again that Figure 9 primarily aims to delineate the conditions for the development of the three main types of deformation accommodation in viscous shear zones. However, the development of fabrics in

microscale will be evidently more complex, as rightly indicated by the reviewer, due to various microscale heterogeneities, which is briefly discussed in this version (Ln: 442-445). We also acknowledge that the present model is applicable to macroscopic analysis of viscous shear zones, and does not account for the possible effect of complex rheology. This limitation is explicated stated in this version (Ln: 368-370, 522-524).

*Terminology of "Ductile shear zones"*

**Comment:** I find the term "ductile shear zones" a little misleading. Maybe, this is a general term in geology, but considering that multiple deformation mechanisms are observed, brittle and viscous, it might be not the appropriate term. The brief explanation of both mechanisms in shear zone (lines 32-36) is good, however, I suggest to generally talk about "shear zones" (maybe "crustal" or "lithospheric shear zones") when describing and discussing the field evidence and the numerical results. Calling them ductile shear zones already implies, that you only consider ductile deformation processes, like grain-size sensitive diffusion creep, non-linear creep, melt interaction (in a simplified formulation) and such. While this might be the case for the natural shear zones, none of the more complex mechanisms are considered in the numerical models. In fact the numerical models are just visco-plastic models with a constant viscosity. The occurrence of the different deformation mechanisms is already discussed in the introduction. However, in the current version, I do not feel like this is enough to justify calling them ductile shear zones. If you do insist on calling them ductile shear zones, I think the numerical models are not appropriate to analyze the natural structures without a discussion on how to link the plastic failure to shear band formation and corresponding strain localization via any kind of microphysical mechanism (such as grain size reduction).

**Response:** We greatly appreciate this discussion by the reviewer. We agree with the reviewer that the use of the term "ductile shear zone" can create confusion. We now use the term "shear zone" in general throughout the manuscript. In places, they are defined as viscous shear zone in the perspective of our rheological modelling. In the title, it is termed as "crustal shear zone", as suggested by the reviewer.

The visco-plastic rheology considered in this study includes a yield criterion, where the post-yield effective viscosity does not remain constant, but undergoes a non-linear decrease with increasing strain, similar to that happens in natural shear zones due to various weakening processes, as rightly indicated by the Reviewer. We mathematically implement such mechanical weakening and viscosity reduction in the modelling (Ln: 277-280).

This version includes detailed microstructural descriptions (Supplementary section S3), as commented by the reviewer, to support our rheological considerations (Ln: 226-236). Based on observations, we report deformation mechanisms and associated

grain scale signatures, such grain size reduction and fabric developments. We greatly appreciate this valuable suggestion by the reviewer.

Minor comments

**Comment-** Line 29: augment … processes

**Response:** Corrected. (Ln: 29)

**Comment:** Line 37: I find "irrespective" a little to hard. Yes on a large scale the evolution of shear zones might be scaled by strain partitioning along macroscopic shear bands. However, the internal deformation mechanisms are important, not only for strain partitioning, but also for strain localization processes.

**Response:** This part of the text is modified to accommodate the reviewer's suggestion (Ln: 38-39).

**Comment:** Line 60: deformation not deformations

**Response:** Corrected. (Ln: 62)

**Comment:** Line 63: … Mair and Abe (2008),…

**Response:** Corrected. (Ln: 65)

**Comment:** Line 67: depends

**Response:** Corrected. (Ln: 69)

**Comment:** Line 76: do you mean … (Rutter et al., 1986), …?

**Response:** Corrected. (Ln: 78)

**Comment:** Line 76: what brittle features? The feedback mechanism shown by Bercovici and Karato (2002) are not brittle features!

Response: This part of description has been modified. We have also changed the sentences to rightly describe the work of Bercovici and Karato (2002), as rightly pointed out by the reviewer. (Ln:85-90)

**Comment:** Line 81: lead

**Response:** Corrected. (Ln: 92)

**Comment:** Line 85: occurs

**Response:** Modified. (Ln: 96)

**Comment:** Line 94: deformation ... occurs ...

**Response:** Modified. (Ln: 105)

**Comment:** Line 162: Definition on how the area is evaluated is unclear.

**Response:** The area calculation is now clearly defined. Thanks. (Ln: 195)

**Comment:** Equation 4: you mean $(1/eta\_v + 1/eta\_p)^{-1}$, correct.

**Response:** Corrected. (Ln: 260)

**Comment:** Equation 6: what is Chi (X)?

**Response:** The term has been defined. (Ln: 277-278)

**Comment:** Line 223: "decreases non-linearly". Is this really the case? Looking at equation (8) the cohesion decreases linearly with increasing strain.

**Response:** We actually meant a nonlinear decrease of the effective viscosity in the post-yield phase, which is shown in a figure included in the Supplementary text (S5). This part of the text is also significantly modified (Ln: 277-280).

**Comment:** Equation (7): This equation does not explain how the flow stress is equated by the yield stress. What is the absolute of the strain rate? Where does the yield function F occur?

**Response:** We have clarified the basis of considering the strain-rate dependent effective viscosity (Ln: 272-275). The absolute of the strain rate and the function F have been appropriately defined (Ln: 277)

**Comment:** Equation (8): Is the pressure that important? I assume, in that case it would simply be the dynamic pressure.

**Response:** P represents the isotropic part of the stress tensor which has been defined mathematically in the main text (Ln: 283).

**Comment:** Line 260: I would not call it bulk viscosity, since this would mean the viscosity of the shear zone. Maybe call it ductile viscosity.

**Response:** Based on the reviewer's suggestions, the term ductile has been removed. Thus we explain the term as initial viscosity, used to represent the viscosity of the shear zone material prior to beginning of shear deformation. Thanks. (Ln: 315)

**Comment:** Table 1: I assume the Cohesion is also the Scaling value for the stress, correct. The correct way is 2.7 10 -14 s-1 and 10 21 Pa s.

**Response:** Yes, the scaling value for stress is the same as that for cohesion. The presentation is modified, as suggested. Thanks for this nice suggestion.

**Comment:** Line 270: Supplementary videos do not work!

**Response:** The video files are uploaded again.

**Comment:** Line 279: ... deformation emerges ...

**Response:** Corrected. (Ln: 336)

**Comment:** Line 415: parentheses missing

**Response:** Corrected. (Ln: 497)

**Comment:** Line 422-426: Little unclear, maybe rephrase a bit. What exactly is an „enormous shear rate enhancement„?

**Response:** This part of the text has been modified. (Ln: 504-507)

Comment: Line 426: Ductile shear zone cannot produce earthquakes, since an earthquake is a brittle event! Even a dramatic viscosity reduction within a shear band and a high viscosity contrast do not produce Earthquakes, but simple a high strain rate event. To obtain an earthquake you need brittle failure. Maybe, shear localization in the ductile regime can trigger an earthquake in shallower brittle regimes.

**Response:** Yes, ductile shear zones cannot produce earthquakes, as they are triggered by brittle (macroscopic fracturing) events. We suggest that shear bands can be a potential location of strain-rate dependent transition from viscous to slip-assisted (shear fracturing) deformation transition when they attain extremely large shear rates. This is how we can explain the mechanics of earthquake generation with great focal depths where viscous deformations are generally the usual mode of deformation. We have modified the text to clarify our main point of discussion. Relevant studies are also cited in the discussion. (Ln: 509-512)

**Comment:** Line 288: what is lambda*?

**Response:** The symbol is now defined (Ln: 166).

**Comment:** Figures 1b and 1c: What are the red squares? The regions of you field studies?

**Response:** Yes, they delineate our field areas. The red squares have been explained the figure caption of the modified version.

**Comment:** Figures 2 and 3: Scale is not very good visible

**Response:** Modified to make the scale clearly visible, as suggested by the reviewer.

**Comment:** Figure 5: What is the color scale in the background? Is it simply a gradient showing the different regimes? In that case, you should leave it out! Maybe call the x-axis like in the text: "areal percentage of S foliage on domains" or similar.

**Response:** The gradient in the background colour is now removed to avoid confusion. The x-axis is defined, as suggested.

**Comment:** Figure 7c(iv): what is the white arrow?

**Response:** The arrow was placed by mistake. It has been corrected in the modified version. Thanks for point this mistake.

**Comment:** Figure 8: How did you obtain the values from the models? Are the interpolated over the particles?

**Response:** We measured the angles from the major axis of strain ellipse (initially circular passive markers set in the initial model domain) at each step of the simulation run. This is clarified in the text (Ln: 329) and figure caption.